# Abrupt perturbation and delayed recovery of the vaginal ecosystem following childbirth

Elizabeth K. Costello [1] ✉, Daniel B. DiGiulio[1], Anna Robaczewska[1], Laura Symul [2], Ronald J. Wong [3], Gary M. Shaw[3], David K. Stevenson[3], Susan P. Holmes[2], Douglas S. Kwon [4,5] & David A. Relman [1,6,7] ✉

The vaginal ecosystem is closely tied to human health and reproductive outcomes, yet its dynamics in the wake of childbirth remain poorly characterized. Here, we profile the vaginal microbiota and cytokine milieu of participants sampled longitudinally throughout pregnancy and for at least one year postpartum. We show that delivery, regardless of mode, is associated with a vaginal pro-inflammatory cytokine response and the loss of *Lactobacillus* dominance. By contrast, neither the progression of gestation nor the approach of labor strongly altered the vaginal ecosystem. At 9.5-months postpartum—the latest timepoint at which cytokines were assessed—elevated inflammation coincided with vaginal bacterial communities that had remained perturbed (highly diverse) from the time of delivery. Time-to-event analysis indicated a one-year postpartum probability of transitioning to *Lactobacillus* dominance of 49.4%. As diversity and inflammation declined during the postpartum period, dominance by *L. crispatus*, the quintessential health-associated commensal, failed to return: its prevalence before, immediately after, and one year after delivery was 41%, 4%, and 9%, respectively. Revisiting our pre-delivery data, we found that a prior live birth was associated with a lower odds of *L. crispatus* dominance in pregnant participants—an outcome modestly tempered by a longer (>18-month) interpregnancy interval. Our results suggest that reproductive history and childbirth in particular remodel the vaginal ecosystem and that the timing and degree of recovery from delivery may help determine the subsequent health of the woman and of future pregnancies.

Delivery marks an abrupt biological transition. Within hours, a pregnancy's unique physiological, immunological, and anatomical supports give way, setting the stage for the activation of lactation, initiation of return to a non-pregnant state, and eventual resumption of cycling and fertility[1]. The vaginal ecosystem, consisting of both the host environment and microbial community, is a key modulator of human health and reproductive outcomes[2]. While this ecosystem appears to be altered in the immediate wake of delivery[3–9], the exact nature, extent, and duration of this perturbation remain unclear. Whether and how the vaginal ecosystem recovers from childbirth could influence health risks in the postpartum period and beyond, including those associated with a subsequent pregnancy[7,10].

[1]Department of Medicine, Stanford University School of Medicine, Stanford, CA 94305, USA. [2]Department of Statistics, Stanford University, Stanford, CA 94305, USA. [3]Department of Pediatrics, Stanford University School of Medicine, Stanford, CA 94305, USA. [4]Ragon Institute of MGH, MIT, and Harvard, Cambridge, MA 02139, USA. [5]Division of Infectious Diseases, Massachusetts General Hospital, Boston, MA 02114, USA. [6]Department of Microbiology & Immunology, Stanford University School of Medicine, Stanford, CA 94305, USA. [7]Section of Infectious Diseases, Veterans Affairs Palo Alto Health Care System, Palo Alto, CA 94304, USA. ✉e-mail: costelle@stanford.edu; relman@stanford.edu

Bacterial communities composed almost entirely of a single species of *Lactobacillus*, and in particular *L. crispatus*, have long been associated with vaginal health. Fermenting lactic acid from host-derived carbohydrates, these communities substantially lower the vaginal pH, which in turn and alongside other defenses, serves to deter microbial invasion and promote homeostasis[2,11–15]. This configuration has been associated with a relatively low risk of acquiring human immunodeficiency virus (HIV)[16] and of delivering preterm[17] and is widely held as the optimal state of the human vaginal ecosystem[2]. *Lactobacillus iners*, while common, is often excluded from this view because of its liminal role in vaginal health and disease[12,13,16,18–20]. Indeed, nonoptimal states, as they are currently construed, are thought to comprise bacterial communities depleted of non-*L. iners Lactobacillus* species, which often encompass diverse sets of strict and facultative anaerobes[21]. While nonoptimal community configurations are also regularly detected in healthy women[13], they are associated with vaginal inflammation and changes to the mucosal barrier[16,22,23], both of which are thought to heighten health risks[2,21].

The vaginal ecosystem is attuned to the human life cycle. In the context of reproductive maturation, circulating estrogen plays a pivotal role[1,24]. Among numerous effects on the vagina, estrogen promotes epithelial thickening and intracellular glycogen accumulation[11,24], and also influences the quantity and quality of the cervical mucus[25]. Glycogen liberated from exfoliated epithelial cells, and the breakdown products of glycogen, are major substrates for vaginal bacteria, especially *Lactobacillus* species[14,26,27]. While it remains unclear whether circulating estrogen, vaginal cell-free glycogen, and *Lactobacillus* levels are precisely synchronized[28–30], optimal states appear to prevail during more estrogenic life phases, emerging briefly in neonates owing to the action of residual maternal estrogen, receding in childhood, reemerging postmenarche, and receding again postmenopause[11,24,31,32]. With respect to pregnancy, estrogen levels peak late in gestation, plummet with the expulsion of the placenta, and remain low while ovarian cycling is suppressed, the timing of which is regulated in part by the intensity and duration of breastfeeding[1,33–35]. Therefore, the postpartum vaginal ecosystem integrates an immediate aftermath of delivery, including an alkaline lochial discharge and any mechanical trauma, with a longer-term, host physiological trajectory involving the course of lactation and return of fertility, phases that vary in length from woman to woman.

A handful of studies have characterized postpartum vaginal bacterial communities, mainly at six or fewer weeks after delivery, and often as extensions of pregnancy-focused work. These studies, while modest in number and scope, have consistently portrayed early postpartum vaginal bacterial communities as being *Lactobacillus*-depleted and more diverse than those of pregnant or not-recently-pregnant women[3–10]. Whether such communities provoke vaginal pro-inflammatory responses in the postpartum period remains an open question (although see refs. 4,8). Our prior work has indicated that nonoptimal vaginal bacterial communities may persist in some women for up to a year after delivery, but only a few women were followed for that length of time[7]. Taken together, these previous findings suggest that delivery may have a relatively large effect on the vaginal microbiota, but that longitudinal follow-up studies are still needed. Without a more comprehensive view of the dynamics of the postpartum vaginal ecosystem, our concept of the natural history of the human microbiome would remain incomplete, as too might our understanding of why postpartum women are at increased risk of developing endometritis[36], acquiring HIV[37], and experiencing adverse birth outcomes associated with a short interpregnancy interval (IPI)[38].

We sought to build upon our earlier work in which we documented a significant post-delivery increase in vaginal bacterial diversity in 25 participants enrolled at Stanford University (SU) and followed throughout pregnancy as part of a larger ongoing study of preterm birth[7]. Here, in participants drawn from the same SU cohort, we focus on the postpartum phase, extending longitudinal sampling of 72 pregnancies to an average of one year after delivery, and following 17 participants over successive gestations. Furthermore, through the use of consistent methods, we were able to incorporate published data from additional pregnant participants enrolled at Stanford and at the University of Alabama at Birmingham (UAB)[17], thus expanding our comparative framework. Our analysis confirms the gestational stability of the vaginal microbiota, details the profound and pervasive effects of delivery on the vaginal ecosystem, and reveals an unsettled postpartum year in which recovery is complicated by the persistence of nonoptimal states. Our findings suggest that birth history significantly influences the vaginal microbiota women bring to their subsequent pregnancies.

## Results

### A longitudinal study of the vaginal ecosystem before and after childbirth

Participants were selected from the Stanford University preterm birth study[7,17,39] on the basis that they had provided vaginal swabs before and after delivery, or over multiple pregnancies. In most cases, study participation continued throughout gestation and into the postpartum period. Using methods consistent with our earlier work[17], 16S rRNA genes were amplified, sequenced, and analyzed from 1,669 unique vaginal swabs self-collected by the participants.

These data were merged at the level of amplicon sequence variant (ASV) with our previously published data from pregnant women enrolled at Stanford and at UAB[17]. The merged dataset comprises 3,848 unique vaginal swabs and represents longitudinal sampling of 82 SU participants over 100 pregnancies and 96 UAB participants over 96 pregnancies. Sixteen SU participants were followed over two pregnancies, and one was followed over three. Supplementary Tables 1 and 2 describe the participants and their pregnancies, respectively, and Supplementary Fig. 1 displays a collection timeline for the vaginal swabs analyzed in this study. Among cases with at least one post-delivery sample available for analysis (*n* = 72 SU pregnancies), 93% delivered at term and the median extent of monthly postpartum sampling was 365 days [interquartile range (IQR) 332-375 days; range 6-790 days]. This subset of pregnancies was generally representative of the SU cohort as a whole (Supplementary Tables 1 and 2, Supplementary Fig. 2a).

Cytokine/chemokine concentrations were evaluated in a series of vaginal swabs (*n* = 198; see Methods for selection criteria) and analyzed in relation to the corresponding, paired, 16S rRNA gene surveys. For *n* = 40 SU pregnancies, cytokines were assessed at approximately 3 weeks before, 6 weeks after, and 9.5 months after delivery, and we refer to these windows as late gestation, early postpartum, and late postpartum, respectively. For two of these pregnancies and 38 additional ones (*n* = 9 SU; *n* = 29 UAB), cytokines were assessed in early gestation (approximately 20 weeks before delivery) and late gestation (Supplementary Fig. 1).

Finally, we point out that the SU and UAB cohorts were geographically, demographically, and clinically distinct (Supplementary Tables 1 and 2)[17]. For example, the UAB cohort had a substantially lower level of nulliparity than the SU cohort (2% versus 42% of pregnancies) because enrollment was from a population of pregnant women at high risk for preterm birth, indicated in most cases by a history of prior preterm birth (Supplementary Table 2).

### Neither the progression of gestation nor the approach of delivery strongly alters the vaginal ecosystem

In our earlier work, we reported on the gestational stability of the vaginal microbiota, which we examined in a group of 40 SU participants[7]. Here, we briefly revisit this relationship (between microbiota and time), given the opportunity to do so in a larger number of pregnancies and in an effort to better understand the dynamics leading up to delivery.

We confirm that while vaginal bacterial diversity (alpha diversity) fluctuated from time to time in pregnant women (Supplementary

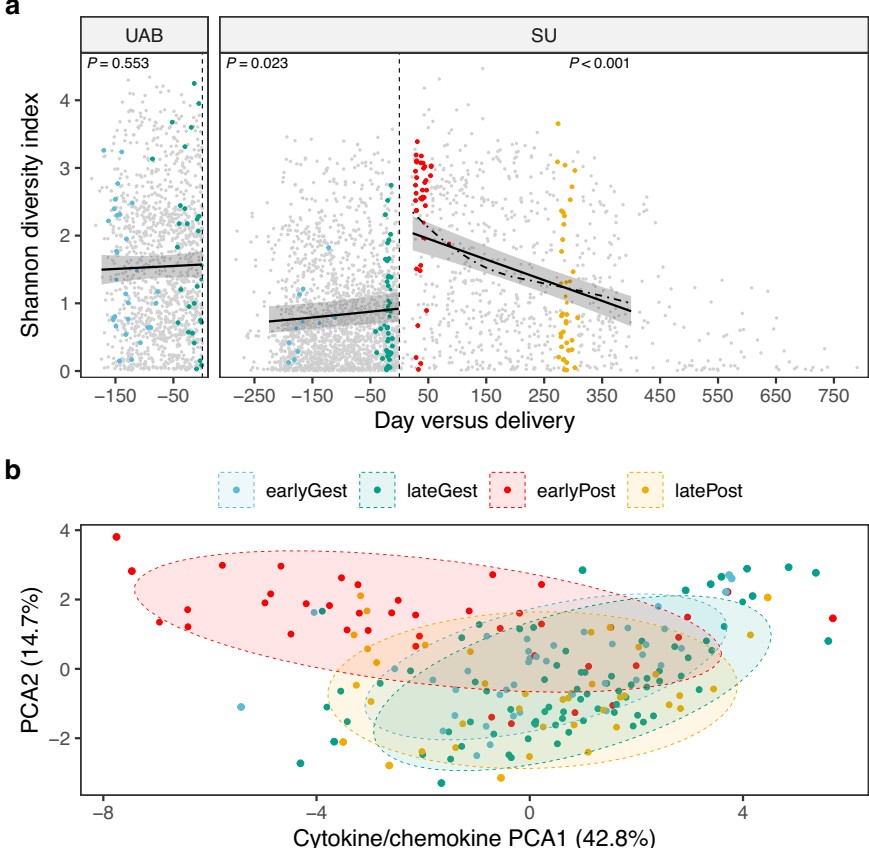

**Fig. 1 | Vaginal ecosystem dynamics before and after childbirth: a 10,000-foot view.** This figure provides a high-level view of the study, as every sample is displayed. In **a**, alpha diversity, measured using the Shannon diversity index, is plotted against day relative to the end of gestation (vertical dashed line). For each point ($n = 3,848$ unique vaginal swabs), 16S rRNA genes were amplified, sequenced, and resolved to amplicon sequence variants (ASVs). For each non-gray point, a paired swab was analyzed for cytokine/chemokine concentrations ($n = 198$ unique vaginal swabs; panel **b**). The data are facetted by cohort (left, UAB cohort; right, SU cohort). Solid lines depict linear mixed-effects regressions and shaded areas represent 95% confidence intervals for the fixed effect term (i.e., day). Per-pregnancy slopes and intercepts were modeled as random effects. Three datasets were modeled, each limited to pregnancies ending in delivery with >1 timepoint available for analysis (UAB before, $n = 93$; SU before, $n = 98$; SU after, $n = 65$). $P$ values correspond to the estimated slopes of the fixed effect term and were evaluated by $t$-test. The rate of postpartum recovery was also explored using local smoothing (loess fit; dot-dashed line). Congruent results were obtained using Faith's phylogenetic diversity as the alpha diversity metric. **b**, Ordination of vaginal samples based on their cytokine/chemokine content. Swabs were collected before and after delivery as highlighted in panel **a**. Principal components analysis (PCA) was applied to the log-transformed concentrations of 16 analytes: IFN-γ, IL-1α, IL-1β, IL-5, IL-6, IL-10, IL-17, IL-21, IL-23, IP-10, ITAC, MIG, MIP-1α, MIP-1β, MIP-3α, and TNF-α. Normal data ellipses (75%) are plotted to aid in visualization. The effect of temporal phase on the cytokine/chemokine content of vaginal samples was tested using a permutational MANOVA based on Canberra distances ($P < 0.001$).

Fig. 2a), these shifts were not strongly associated with the progression of gestation (Fig. 1a and Supplementary Fig. 2b). Similarly, while we observed a significant degree of temporal turnover in vaginal bacterial community composition (beta diversity) within women (Supplementary Fig. 2c), when we examined a relatively large time span (around 18 weeks of gestation), we found no evidence of a concerted shift in community composition (Supplementary Fig. 2d), nor of any differentially abundant ASVs (Supplementary Fig. 2e), nor of any coherent change in the vaginal cytokine milieu (Fig. 1b and Supplementary Fig. 2f). In a third of all pregnancies, the identity of the top ASV (the most abundant ASV) remained the same throughout gestation and, in each of these cases, was a *Lactobacillus* ASV (Supplementary Fig. 2g), underscoring the notion that maximum stability is associated with *Lactobacillus* dominance. Taken together, these results are consistent with a scenario in which pregnancy-associated vaginal bacterial communities display modest temporal fluctuations in diversity and composition that are not strongly directed by the advancing pregnancy.

In the final few weeks of gestation, maternal physiology transitions from a state of pregnancy maintenance to a state of impending labor[1,40]. For pregnancies in our study with spontaneous labor onset (Supplementary Table 2), we considered whether changes in the

vaginal microbiota may have heralded this transition. In most cases, weekly observations were available for the month preceding delivery (Supplementary Fig. 1). However, we detected no significant trend in vaginal bacterial diversity over the final five weeks of gestation in either cohort (Supplementary Fig. 2h), nor, for this timeframe, did we detect any difference in the average level of diversity compared to pregnancies without spontaneous labor onset. We conclude that vaginal bacterial diversity was largely unresponsive to host physiological changes preceding the spontaneous onset of labor. Although our results suggest little in the way of a pre-delivery disturbance, we note that recent studies have detected a subtle compositional shift occurring after labor onset[41] or membrane rupture[42] but before birth, an interval we did not sample. The exact starting point of any delivery-associated disturbance remains unknown, as most studies, including this one, pause sampling at and for several weeks after delivery.

## Cohort-associated differences in the diversity and composition of the vaginal microbiota
Cohort-associated differences in pregnant women, if present, might not persist after delivery. While we detected little change associated with the progression of gestation in either cohort, we observed clear

differences between cohorts, regardless of gestational stage. Vaginal bacterial diversity (Fig. 1a and Supplementary Fig. 2b), temporal turnover in community composition (Supplementary Fig. 2c), and instability in top ASV identity (Supplementary Fig. 2g) were each significantly higher in the UAB cohort when compared with the SU cohort (see Supplementary Fig. 2 caption for results of statistical tests).

Furthermore, samples from pregnant SU participants were more likely to be dominated by a single ASV identified as *L. gasseri*, *L. crispatus*, or *Bifidobacterium breve/longum*, and these ASVs had higher average gestational frequencies in the SU cohort than in the UAB cohort (Supplementary Fig. 3; average gestational frequency is the mean frequency of an ASV across all samples from the same gestation[17]). Samples from UAB participants were more likely to be non-dominated, with no single ASV in the majority (Supplementary Fig. 3). Higher average gestational frequencies of asv12 [*Ca.* Lachnocurva vaginae (BVAB1)][43], asv9 (*Megasphaera lornae*)[44], asv18 (uncultivated *Prevotella* sp.)[45], and asv23 (*Sneathia amnii*) most clearly distinguished UAB from SU participants (Supplementary Fig. 3). In short, UAB participants were more likely than SU participants to harbor vaginal microbiota that might be considered nonoptimal. Although the cohort-based differences we observe here are consistent with those reported for race-stratified comparisons within cohorts[13,46–48], we were unable to disentangle race from other factors that varied systematically between our two cohorts (such as treatment, nulliparity, and geographic location; Supplementary Tables 1 and 2).

We surmised that vaginal inflammation might accompany these nonoptimal vaginal bacterial communities[49]. When we compared late gestational vaginal cytokine concentrations between the two cohorts, we found that UAB participants had significantly higher levels of interleukin 10 (IL-10; two-sided Wilcoxon rank sum test, $P < 0.01$) and IL-17 ($P < 0.01$), and lower levels of monokine induced by interferon-γ (MIG; $P < 0.001$) and interferon-γ induced protein 10 (IP-10; $P < 0.0001$). However, these results might also reflect cohort-associated differences in treatment. UAB participants received intramuscular (IM) 17α-hydroxyprogesterone caproate (17-OHPC) throughout their pregnancies[17] and this treatment has been linked to alterations in the late gestational vaginal cytokine milieu[50]. By contrast, a related therapy, vaginal progesterone, has been shown to have little impact on the diversity and composition of the vaginal microbiota[51].

While a few of these cohort-associated differences in vaginal microbiota composition have been described in earlier work[17], herein, they also provide a framework for probing whether (or, to what degree) SU participants' postpartum vaginal bacterial communities retain cohort-consistent features (i.e., remain SU-like), adopt UAB-like features, or represent a novel state.

## Delivery profoundly remodels the vaginal microbiota and cytokine milieu

Delivery was associated with a significant increase in vaginal bacterial diversity (Fig. 1a and Fig. 2a), elevating the effective number of ASVs from a median of two among the last gestational samples (IQR 1-3 ASVs) to 13 among the first postpartum samples (IQR 5-18 ASVs). Eighty percent of deliveries were accompanied by an increase in vaginal bacterial diversity; in 63% of cases, bacterial diversity in the first postpartum sample exceeded that of any sample collected before delivery. Anecdotally, in two cases of miscarriage (at 11 and 13 weeks), we did not observe this type of disturbance.

Delivery also interrupted the temporal dependence of the time-series, as the level of diversity in the first postpartum sample was only weakly correlated with that of the last gestational sample (Spearman's rho = 0.15, $P = 0.23$, $n = 70$ deliveries) or with the average gestational diversity (Spearman's rho = 0.10, $P = 0.42$). Neither the level of diversity in the first postpartum sample, nor the degree to which it changed with delivery was predicted by any delivery-associated variable that we examined, which included maternal age at delivery, prior nulliparity,

labor onset type, membrane rupture type, delivery mode (indicated in Fig. 2a), group B *Streptococcus* (GBS) status, peripartum antibiotic exposure, gestational day of delivery, and the sex of the baby.

The effect of delivery on the vaginal microbiota may best be understood as a loss of dominance (Fig. 2b). While rank-based testing (Supplementary Table 3), differential (relative) abundance testing (Supplementary Fig. 4a), and phylogeny-informed sparse discriminant analysis (Supplementary Fig. 4b) each highlighted acute declines in the relative abundance of *Lactobacillus* species, other dominant taxa, while uncommon, were also diminished (Fig. 2c). It was unusual for a particular ASV to remain in the majority across delivery (Fig. 2b); this occurred in only 5/59 cases (8%), including three cases for asv1 (*L. iners*) and two cases for asv2 (*L. crispatus*) (Fig. 2c). True to its dual nature, *L. iners* appeared to be sensitive to delivery, while also faring better than other *Lactobacillus* species in the early postpartum environment (Fig. 2c). Finally, broadening our view, we found that monthly-scale turnover in the composition of the vaginal microbiota for the interval spanning delivery far exceeded that of any other interval in our study (Fig. 2d).

Delivery was also accompanied by a strong shift in the vaginal cytokine milieu, characterized by increased levels of IL-10, IL-17, IL-23, IL-6, macrophage inflammatory protein-1α (MIP-1α), MIP-1β, MIP-3α and tumor necrosis factor-α (TNF-α), and decreased levels of IL-1α and IP-10, which occurred in vaginal and cesarean deliveries alike (Fig. 1b and Supplementary Fig. 5a, b). These changes are indicative of increased vaginal inflammation in the immediate wake of delivery, occurring in response to (or alongside) the increased levels of vaginal bacterial diversity. Interestingly, in two participants who retained (or quickly recovered) high levels of asv2 (*L. crispatus*) (Fig. 2c), this spike in vaginal inflammation was not observed (Supplementary Fig. 5b).

Taken together, these results suggest that childbirth has the capacity to dramatically remodel the vaginal ecosystem, and, given our current understanding of what constitutes vaginal health, not for the better. Yet, the pervasiveness of the perturbation, and its insensitivity to the pre-delivery state or to delivery-associated events, points toward drivers intrinsic (i.e., nearly universal) to delivery—the abrupt and dramatic changes that occur in each host.

## Early postpartum vaginal ecosystem: a cohort-consistent, nonoptimal state that persists in some women

Diverse, early postpartum vaginal bacterial communities were composed of a wide array of facultative and obligate anaerobes (Fig. 3a, Supplementary Fig. 4, and Supplementary Table 3). (Here, we define "diverse" by a Shannon diversity index >2.) Among the most abundant constituents of these communities were members of the genera *Prevotella*, *Anaerococcus*, *Finegoldia*, *Peptoniphilus*, and *Streptococcus*. This compositional profile is reminiscent of McKinnon et al.'s (2019) community state type (CST) IVC[21] and France et al.'s (2020) CST IV-C0[52] (described by the authors as even with moderate amounts of *Prevotella*). We found that the composition of diverse postpartum communities remained largely SU-like (Fig. 3a and Supplementary Fig. 6). Features characteristic of the UAB cohort—for example, relatively high frequencies of asv9 (*Megasphaera lornae*), asv12 [*Ca.* Lachnocurva vaginae (BVAB1)], asv18 (uncultivated *Prevotella* sp.), and asv23 (*Sneathia amnii*)—were not acquired postpartum by the SU cohort (Fig. 3a and Supplementary Fig. 6a, b), nor did delivery usher in a completely novel set of diverse taxa (Supplementary Fig. 6c). Of note, two SU participants exhibited high levels of asv9 (*Megasphaera lornae*) throughout pregnancy; in each case, asv9 was diminished after delivery (Supplementary Fig. 6b), highlighting that some taxa which have been associated with adverse health outcomes, such as *Megasphaera*[44], might also be susceptible to this disturbance.

Vaginal inflammation and bacterial diversity waned considerably over the months following delivery (Fig. 1 and Supplementary Fig. 5). Given the strong temporal trends in our data, we sought to identify

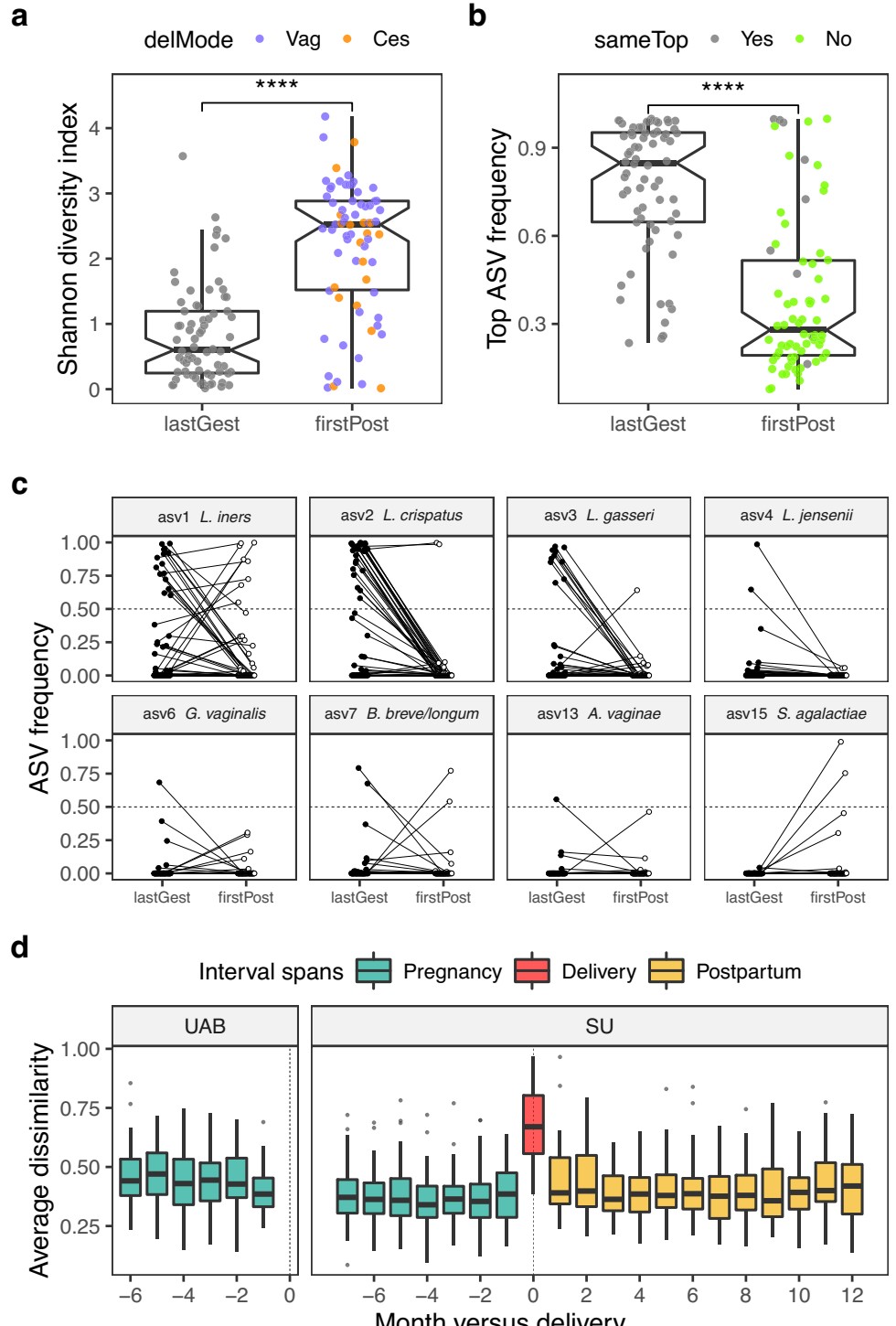

taxon-cytokine pairs whose dynamics were tightly coordinated across the three timepoints (~ 3 weeks before, 6 weeks after, and 9.5 months after delivery). We visualized these relationships using a repeated measures correlation-based network (Fig. 3b), which revealed a core set of elements exhibiting concerted behavior within participants. This nexus—the nodes connected by purple edges in Fig. 3b—defines the early postpartum ecosystem state for the SU cohort. It depicts an interface between vaginal microbiota and the local immune response—a set of elements that rise and fall together—in the context of delivery.

By the tenth postpartum month, the vaginal ecosystem had, to a large extent, recovered (Fig. 1). By that time, for example, seven of the 10 cytokines that responded to delivery had returned to within pre-

delivery levels (Supplementary Fig. 5a). However, we also noticed that late postpartum cytokine profiles for several women clustered with the early postpartum ones (Supplementary Fig. 5b). Overall, three cytokines remained elevated at the late postpartum timepoint (IL-10, IL-23, and MIP-1β; Supplementary Fig. 5a), as did the Shannon diversity index (two-sided Wilcoxon signed rank test, $P = 0.008$). It seemed possible that recovery was incomplete or delayed in at least a few women.

To explore this possibility, we analyzed late postpartum samples using an ordination method (CCA) that constrained variation in taxon frequencies to that related to cytokine levels (Fig. 3c). This identified 12 participants who displayed a late postpartum vaginal milieu reminiscent of the early postpartum state (Fig. 3c; points located in the upper

**Fig. 2 | Delivery is accompanied by a strong shift in the diversity and composition of the vaginal microbiota that is unsurpassed across the reproductive cycle.** Panels **a** to **c** focus on pairs of samples immediately flanking delivery (SU cohort, $n = 70$ pregnancies). Pairs consist of a last gestational sample (lastGest) and a first postpartum sample (firstPost). These samples were collected a median of 7 days before (IQR 5-10) and 38 days after delivery (IQR 29-47), respectively, and 44 days apart (IQR 35-61). In **a**, the Shannon diversity index, a measure of alpha diversity, is plotted for each pair. For the postpartum samples, color corresponds to the mode of delivery (vaginal or cesarean). ****, $P = 2e-10$, two-sided Wilcoxon signed rank test. In **b**, the relative abundance of each sample's most abundant ASV (top ASV) is plotted, with postpartum color indicating a change in the identity of the top ASV with delivery. ****, $P = 3e-10$, two-sided Wilcoxon signed rank test. Boxplots depict the median, approximate 95% CI (notches), IQR (hinges), and most

extreme values within 1.5 * IQR of the hinges (whiskers). **c**, Relative abundance of individual ASVs before and after delivery. Selected for display are ASVs with supermajority status in any lastGest or firstPost sample. Panel **d** quantifies and contextualizes the observed levels of delivery-associated compositional instability. In **d**, the average within-participant Bray-Curtis dissimilarity, a measure of beta diversity, is plotted against month versus delivery. Pairwise dissimilarities between same-pregnancy samples collected 3.5 to 9 weeks apart were binned to the start month (the earliest month in the pair) and averaged. Intervals spanning delivery were manually placed at month 0 before averaging. Dissimilarities were calculated using fourth root transformed count data to moderate the influence of extremely dominant ASVs. Boxplots depict the median, IQR (hinges), and most extreme values within 1.5 * IQR of the hinges (whiskers). Results were robust to choice of alpha- and beta-diversity metrics.

right quadrant). We found that these participants were not simply experiencing an arbitrary, nonoptimal day: their trajectories from delivery up to the late postpartum timepoint exhibited persistently high bacterial diversity (Fig. 3d). This was not predicted by a higher pre-delivery setpoint: on average, these participants were no more diverse than others during gestation (two-sided Wilcoxon rank sum test, $P = 0.14$). In summary, our results indicate that for nearly a third of the cases analyzed for both cytokines and microbiota, delivery-associated nonoptimal features persisted into the tenth postpartum month.

### Waning postpartum diversity belies an expanded cast of quasi-dominant taxa and weak recovery of L. crispatus

Delivery was associated with a sharp decline in the prevalence of communities dominated by a *Lactobacillus* ASV (i.e., any ASV assigned to the genus *Lactobacillus*) (Fig. 4a). With bacterial diversity trending downward over the postpartum year (Fig. 1a), we wondered if *Lactobacillus* levels rebounded in kind. Using Kaplan-Meier analyses and defining dominance at an ASV frequency > 0.70[17], we found that the one-year postpartum probability of transitioning to *Lactobacillus* dominance by any species was 49.4% (95% confidence interval (CI) [33.6%, 61.5%]; $n = 58$ at-risk cases, 86.2% of which experienced *Lactobacillus* dominance prior to delivery). Indeed, as postpartum Shannon diversity recovered to near pre-delivery levels (Fig. 1a; two-sided Wilcoxon signed rank test, $P = 0.035$, predelivery versus one-year postpartum), the prevalence of *Lactobacillus* dominance by any species remained low (Fig. 4a). Top ASVs from the genera *Bifidobacterium*, *Streptococcus*, *Gardnerella* and *Atopobium*, which increased in prevalence over the postpartum year and have the capacity to attain relatively high frequencies (Fig. 4a and Supplementary Fig. 7), likely contributed to the overall waning diversity.

We evaluated a set of predictors for associations with postpartum time-to-*Lactobacillus* dominance using univariate Cox regression models. The response variable in these models was the same time-to-event object as described in the prior paragraph. We found that the probability of *Lactobacillus* dominance was higher among those initiating contraception (any type; Supplementary Table 4; typically, prior to 90 days postpartum) compared to those who never initiated (Supplementary Fig. 8a; 90-day landmarked analysis; hazard ratio (HR) 4.82; 95% CI [1.07, 21.71]; $P = 0.04$; $n = 25$ at risk cases). Using time-dependent analyses, we also tested for an association with the resumption of menses (HR 0.54; 95% CI [0.14, 2.16]; $P = 0.39$; $n = 35$ intervals in 25 cases) and the cessation of lactation (HR 2.87; 95% CI [0.96, 8.62]; $P = 0.06$; $n = 41$ intervals in 29 cases), noting a positive association for the latter event. Finally, we tested each of the following predictors, finding a poor association for each (all $P$ values > 0.1): gestational mean *Lactobacillus* frequency, gestational mean Shannon diversity index, maternal age at delivery, prior nulliparity, labor onset type, membrane rupture type, delivery mode, GBS status, peripartum antibiotic exposure, gestational day of delivery, and the sex of the baby. Taken together, our findings suggest that

postpartum recovery of *Lactobacillus* dominance occurs at a relatively slow rate, but might be hastened by hormonal, behavioral, and/or other factors related to the initiation of contraception or the termination of breastfeeding.

We also investigated the incidence of dominance at the level of individual *Lactobacillus* species (Fig. 4b, c), prompted in part by the observation that *L. iners* seemed more tolerant of the early postpartum environment than were other *Lactobacillus* species (Fig. 2c and Supplementary Fig. 8b). We also included *Bifidobacterium* in this analysis because it did not track with late postpartum inflammation (Fig. 3c) and is capable of inducing a relatively low vaginal pH[52], neither of which is true of, e.g., *Streptococcus*. (In other words, we surmised that *Bifidobacterium* might not be harmful.) We found that dominance (ASV frequency > 0.70) by *L. iners*, *Bifidobacterium*, or *L. gasseri* emerged relatively early in the postpartum year (Fig. 4c) and achieved overall incidences nearly as high (*L. iners*, *L. gasseri*) or higher (*Bifidobacterium*) than that observed prior to delivery (Fig. 4a, b). The situation was starkly different for *L. crispatus*, the most common dominant taxon in the SU cohort prior to delivery (Fig. 4a, b). We observed only five cases of postpartum transitions to *L. crispatus* dominance, of which, at least one was not sustained (Fig. 4c, d). Indeed, the prevalence of *L. crispatus* dominance clearly failed to rebound over the postpartum year (Fig. 4a, b). Despite the paucity of events to examine, we remained curious about the host- and community-context in which *L. crispatus* transitioned to high frequency. We noted that in one participant, a transition occurred in the second postpartum year (Fig. 4c, d) and that this transition was preceded by periods in which *Streptococcus* and then *Bifidobacterium* were highly abundant (Fig. 4e). Our results suggest that when *L. crispatus* is absent or diminished after delivery—whether or not it was extirpated or driven to low abundance by delivery—its postpartum introduction and/or expansion is disfavored or delayed relative to other species.

We found that the vaginal ecosystem changed over the postpartum year—recovered, even, with waning levels of diversity and inflammation—in ways shaped by host and microbial factors. However, the cohort did not return to the same state-of-affairs manifested prior to delivery; in particular, the prevalence of *L. crispatus* dominance remained low. Other community states appeared to fill the void, including those in which non-*Lactobacillus* species (e.g., *Bifidobacterium*) were quasi-dominant, inviting the question of whether such states might signal or shape the ecosystem's receptivity to future *Lactobacillus*, particularly *L. crispatus*, invasion.

### Past is prologue: A history of prior live birth is reflected in the vaginal microbiota of pregnant women

Prompted by two observations regarding *L. crispatus* dominance—its poor postpartum recovery in our SU cohort and its relatively low prevalence in our 98%-parous UAB cohort (Fig. 4a)—we hypothesized that the vaginal microbiota women bring to their current pregnancies has been shaped, in part, by past experiences of childbirth.

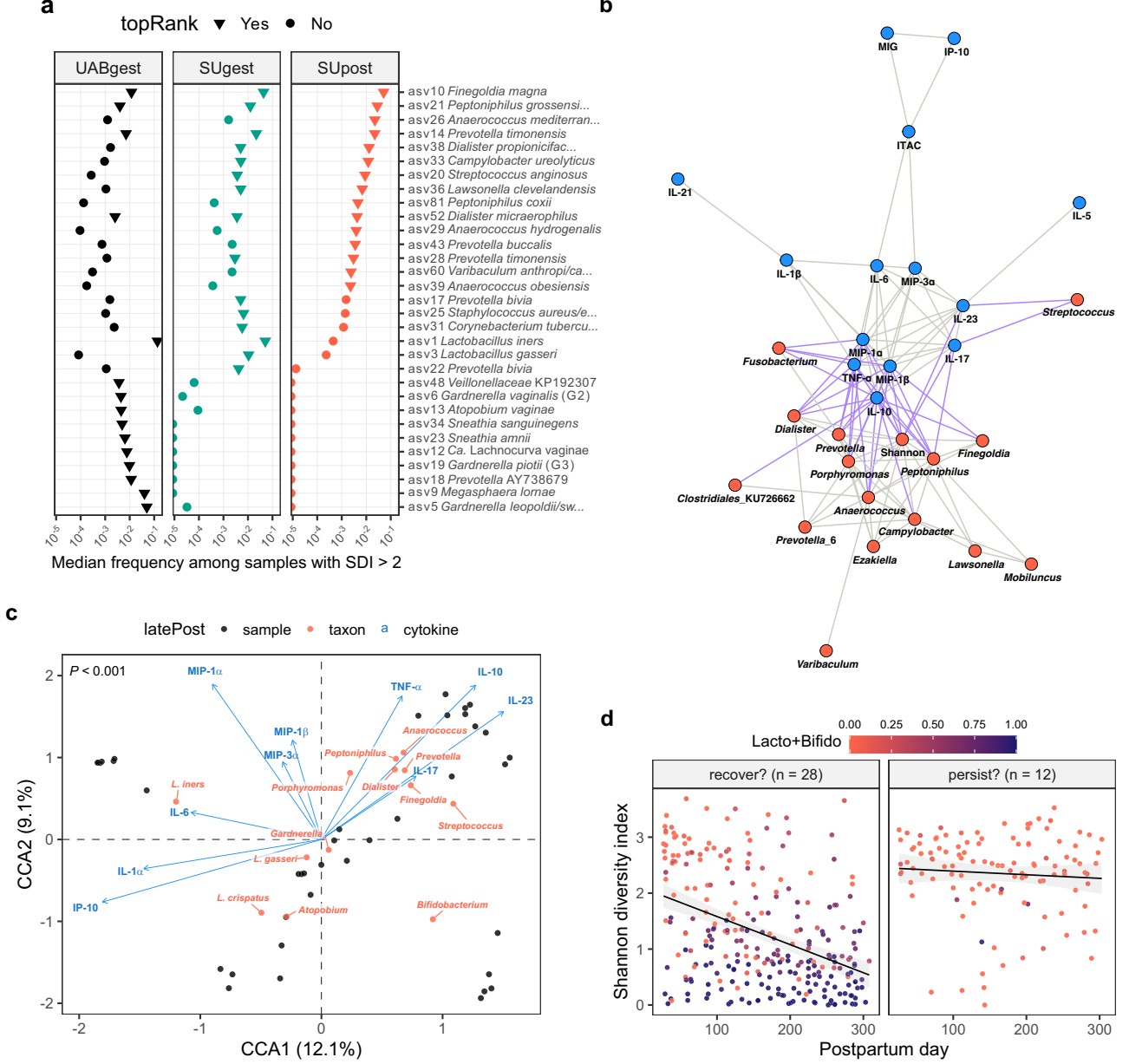

**Fig. 3 | An early postpartum nexus of diverse taxa and pro-inflammatory cytokines that persists in some women >9 months after delivery. a** Median frequencies of 31 ASVs in all high-diversity samples (Shannon diversity index (SDI) > 2) collected before or after delivery. Pregnancies were partitioned by cohort and status (UABgest, $n = 75$; SUgest, $n = 31$; SUpost, $n = 54$, delivery up to four months postpartum). The ASVs represent the union of the top 15 ASVs within each partition (triangles) and are ordered along the y-axis by median frequency in SUpost (decreasing; first 21) or UABgest (increasing; last 10). **b** Correlation network depicting relationships among taxa (red nodes) and cytokines (blue nodes) in 40 pregnancies, each sampled ~3 weeks before, ~6 weeks after, and ~9.5 months after delivery. Edges represent significant, positive, repeated measures correlations among node variables (Benjamini-Hochberg-adjusted $P < 0.001$; $r_{rm} > 0.35$). Mixed edges (taxon-cytokine) are highlighted in purple. The frequencies of 22 bacterial genera and four *Lactobacillus* species, as well as the SDI, were input as taxa variables. Cytokine variables were the $\log_{10}$-transformed concentrations of the 16 cytokines/chemokines depicted in Supplementary Fig. 5a. Isolated nodes were removed from the graph. **c** Constrained correspondence analysis (CCA) of samples collected ~9.5 months after delivery ($n = 40$ pregnancies). The community data matrix contained the frequencies of 23 taxa: 20 bacterial genera and three *Lactobacillus* species. The constraining matrix consisted of the $\log_{10}$-transformed concentrations of the 10 cytokines/chemokines that responded to delivery (Supplementary Fig. 5a). The ordination tri-plot depicts samples (black), taxa (red), and cytokines (blue). Forty percent of the inertia (variation) in taxa was constrained by (related to) the cytokines. The *P* value corresponds to an ANOVA-like permutation test for the joint effect of the constraints. To minimize crowding, several taxa were omitted from the upper right quadrant. **d** Shannon diversity index plotted against postpartum day for the pregnancies depicted in panel **c**. Samples up to and including the latePost sample are displayed. Pregnancies were divided into two groups: those appearing in the upper right quadrant of panel **c** (CCA1 > 0.5 and CCA2 > 0; $n = 12$; right facet) or not ($n = 28$; left facet). Color indicates the summed frequencies of *Lactobacillus* and *Bifidobacterium*. Linear fits are shown for the two groups.

---

Detailed reproductive histories were available for the participants in our study. Setting aside our postpartum data and focusing solely on samples collected before delivery, we found that pregnant women with any history of a prior live birth had a lower odds of harboring majority-*L. crispatus* vaginal bacterial communities than did pregnant women with no such history (OR 0.14; 95% CI [0.06, 0.32]; $P < 0.001$; Fig. 5a). For all other states, the odds were higher—with the topmost being for the majority-*L. gasseri* state (Fig. 5a). Pregnant women with a

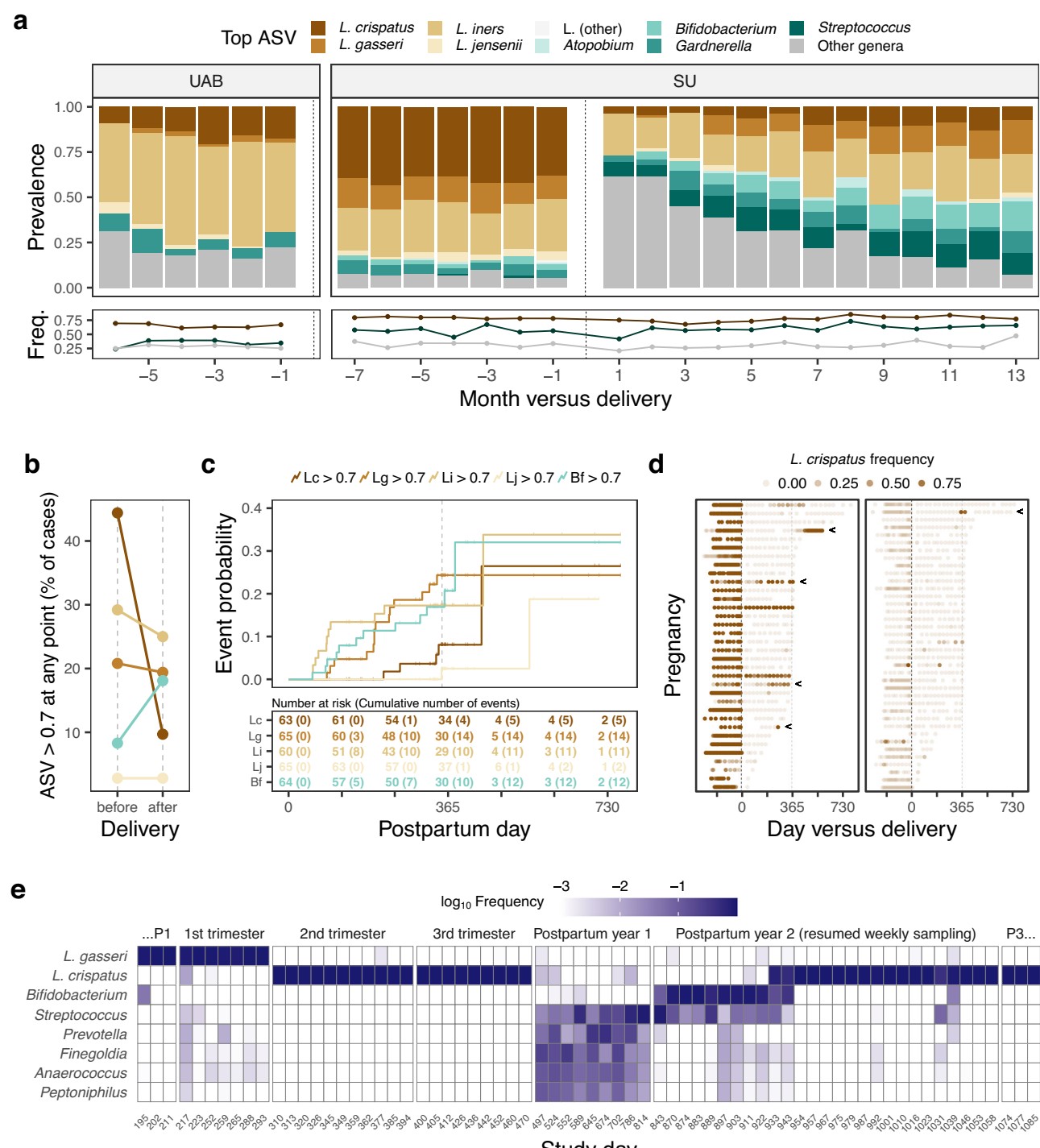

history of prior live birth also had a higher level of vaginal bacterial diversity on average over the course of gestation (two-sided Wilcoxon rank sum test, $P = 0.03$), including in several cases in which the prior gestation was enrolled in our study (Fig. 5b). We note that these comparisons were not possible within the UAB cohort because nearly all UAB participants had a history of prior live birth (Supplementary Table 2).

Finally, we predicted that if *L. crispatus* had a relatively slow rate of postpartum recovery to high abundance (e.g., as depicted in Fig. 4e), then we might observe an effect of birth spacing on its prevalence. Indeed, we found that *L. crispatus*-predominant gestations were slightly more common among women whose prior live birth had

occurred >18 months before the conception of the current pregnancy, compared to those with a more recent history (≤18 months; Fig. 5c; two-sided Fisher's exact test, $P = 0.04$, SU and UAB cohorts combined). On the other hand, *L. crispatus*-predominant gestations were no more common among women with one versus more than one prior live birth, suggesting a lack of dose response beyond the first child.

Taken together, our results suggest that reproductive history shapes the vaginal ecosystem. We posit that the profound perturbation that occurs at delivery in most women, and the potential absence of factors favoring recovery over the postpartum year, act together as a major transformative event in the reproductive cycle. While appreciable recovery takes place after delivery, it appears that the most

**Fig. 4 | A postpartum year characterized by an expanded cast of dominant taxa and poor recovery of *L. crispatus*. a** Stacked bar plots depicting the proportion of samples in which the most abundant ASV (top ASV) belonged to the given taxa. Among same-pregnancy, same-month samples, the sample closest to delivery is shown. Plotted below are the average frequencies of the top ASVs binned to *Lactobacillus* (high; brown); *Atopobium*, *Bifidobacterium*, *Gardnerella*, or *Streptococcus* (intermediate; teal); or other genera (low; gray). Panel **a** includes all study pregnancies that ended in delivery (*n* = 194). **b** For the given timeframe, percentage of pregnancies with at least one sample dominated (frequency > 0.7) by an ASV classified as *Lactobacillus crispatus* (Lc), *L. gasseri* (Lg), *L. iners* (Li), *L. jensenii* (Lj), or *Bifidobacterium* (Bf) (color as in panels **a** and **c**; *n* = 72 SU pregnancies with postpartum follow-up). **c** Postpartum time-to-dominance curves for the five taxa depicted in panel **b**. Samples were defined as "dominated" if they contained an ASV classified to the focal taxon at a frequency > 0.7, and cases "at risk" if they were not dominated at the first postpartum sample and a follow-up sample was available for analysis. For each at-risk case, the first day on which the frequency of the ASV exceeded 0.7 (event) or, barring an event, the last sampled day (censor) was recorded. The curves are Kaplan-Meier step functions with tick marks indicating

censored cases. **d** Timelines depicting *L. crispatus* frequency before and after delivery. On the left are gestations in which *L. crispatus* tended to be more abundant (*n* = 34), including two in which dominance was sustained across (or quickly recovered after) delivery. All others are shown at right (*n* = 38). Arrows mark the five timelines in which events (depicted in panel **c**) occurred. Panel **d** highlights the infrequency with which *L. crispatus* dominance establishes (right) or re-establishes (left) in the period following delivery. **e** Temporal dynamics in an individual participant. Heatmap displaying the $\log_{10}$-transformed frequencies of the most abundant taxa in samples collected throughout pregnancy (1st through 3rd trimesters) and the subsequent interpregnancy interval (Postpartum years 1 and 2). This was the participant's second enrolled pregnancy. Also displayed are the last three consecutive samples (…P1) collected before the start of the focal pregnancy and the first three consecutive samples (P3…) collected after the start of the next pregnancy. The timeseries depicts two transitions to *L. crispatus*-dominance. The second was sustained until the delivery of the next pregnancy. Study day is relative to the participant's first sample, which was collected prior to the start of her first enrolled pregnancy, which ended in miscarriage. Prior to $\log_{10}$-transformation, frequencies <0.001 were set to 0.001.

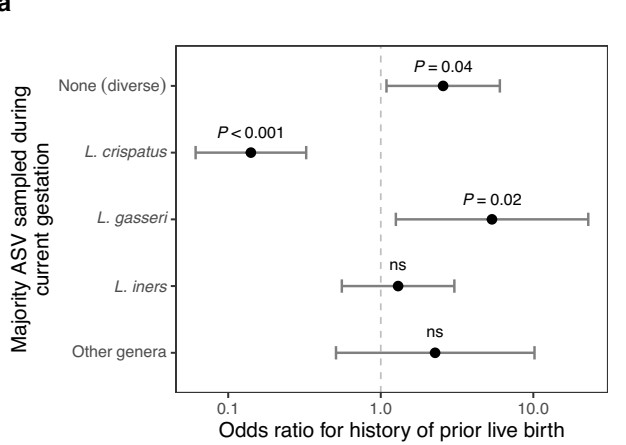
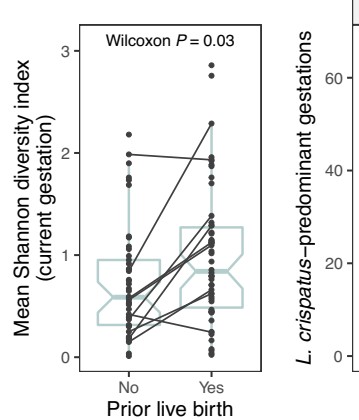
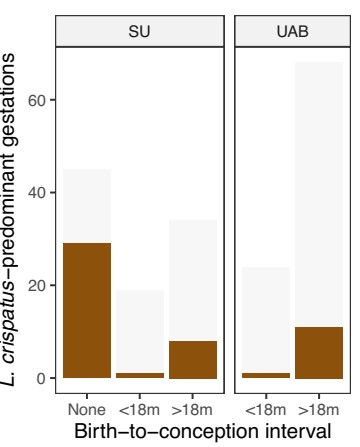

**Fig. 5 | Lower odds of majority-*L. crispatus* vaginal microbiota in pregnant women with a history of prior live birth. a** Associations between vaginal bacterial community states (outcomes) and a history of prior live birth (exposure) in *n* = 98 SU gestations (*n* = 53 with a history of prior live birth). Six states were defined using features of the top ASV (majority status and taxonomic identity) and modeled separately as dichotomous outcomes (one-versus-rest). Wald 95% CIs were calculated using a clustered covariance matrix which accounts for correlation among longitudinal samples. Differences in sampling effort were addressed via repeated random draws of six samples per gestation (100 iterations; odds ratios, CIs, and *P* values represent means). Similar results were obtained for crude (shown) and maternal age- and race-adjusted models. Results for the uncommon *L. jensenii*-dominated state (top ASV frequency > 0.5 and identified as *L. jensenii*) were unstable and are not shown. **b** Average Shannon diversity index for SU gestations stratified by a history of prior live birth. The *P* value corresponds to a two-sided

Wilcoxon rank sum test. Black lines connect gestations that occurred in the same person (*n* = 9 participants with no history of prior live birth at first enrollment). Boxplots depict the median, approximate 95% CI (notches), IQR (hinges), and most extreme values within 1.5 * IQR of the hinges (whiskers). **c** Number of *L. crispatus*-predominant gestations (brown bars) stratified by cohort and birth-to-conception interval (time between conception of the current gestation and the prior live birth). Birth-to-conception intervals were binned to ≤18 or >18 months, the longer of which was associated with a higher incidence of *L. crispatus*-predominance (two-sided Fisher's exact test, *P* = 0.04, SU and UAB cohorts combined). Those lacking an interval (None; because they had no history of prior live birth) are shown for reference. *L. crispatus* predominance was defined as a species-level average gestational frequency exceeding 0.5. Light gray bars indicate the total number of gestations.

persistent types of change, for example, the reduced incidence of optimal *L. crispatus* dominance, have the capacity to reverberate into the next pregnancy. Our findings hint that the status of the vaginal ecosystem might be considered among those factors indicating or underpinning the lower risk of adverse maternal and infant outcomes associated with adequate birth spacing.

## Discussion

The vaginal mucosa is a hormonally- and immunologically-responsive tissue that reflects and balances the demands of human reproduction and pathogen resistance[1,53,54]. It is optimally able to meet these demands when colonized by an indigenous microbiota that produces lactic acid, reduces the vaginal pH, preserves barrier integrity, and minimizes inflammation—a scenario associated with non-*L. iners*

*Lactobacillus* dominance and a decreased risk of adverse sexual and reproductive outcomes[2]. In the present study, we characterized the vaginal microbiota and cytokine milieu of participants enrolled early in pregnancy and sampled longitudinally throughout gestation (SU and UAB cohorts) and, in many cases, for one year postpartum (SU cohort). Our main finding is that the configuration of the vaginal ecosystem depends on whether, and how long ago, a woman has given birth. We traced this relationship back to changes that occurred, not with advancing pregnancy, but with delivery, which was associated with a strong perturbation of the vaginal microbiota and cytokine milieu. We documented variation among women in the rate and pattern of postpartum recovery, and showed that *L. crispatus* dominance, the quintessential health-associated state, was less prevalent among postpartum and parous hosts compared to those with no recent

history of childbirth. We expect these findings to inform the design of future research, broaden our perspectives on the natural history of the human microbiome, and prompt new strategies for monitoring and supporting vaginal health.

In agreement with our earlier work[7], we found that the vaginal ecosystem was relatively stable to the progression of gestation. Temporal variation within pregnant women did not amount to a concerted shift toward a distinct, late-pregnancy-associated state in either cohort. Our findings are consistent with prior longitudinal studies of the pregnancy-associated vaginal cytokine milieu[4,55] and microbiota[56–58] and with studies suggesting that compositional states wholly unique to pregnancy have not been identified[57,59]. Throughout gestation, in lieu of cycling and menses, which regularly perturb the vaginal ecosystem[54,60,61], circulating estrogen concentrations rise steadily, increasing by two to three orders of magnitude[1]. These hormonal dynamics, in turn, orchestrate changes in the vaginal mucosa[53,54]. As pregnancy advances, the vascularity of the tissue, the thickness of the epithelium, glycogen content of the epithelial cells, and cell-shedding into the vaginal lumen increase, with peak proliferation and development attained mid-pregnancy and sustained until delivery[53]. These conditions are thought to favor *Lactobacillus* dominance. A weakness of our study is that it did not capture the earliest stages of pregnancy. In recent work, Serrano et al. (2019) detected higher levels of vaginal bacterial diversity and a lower prevalence of *L. iners* dominance in first- versus subsequent-trimester samples from women of African ancestry, but not in those from women of non-African ancestry (all enrolled from clinics in Virginia, USA)[58]. Unfortunately, because we obtained few first-trimester samples, especially from our UAB cohort in which most participants self-identified as Black, we could not address whether similar early-pregnancy shifts occurred in our study, nor could we have disentangled ancestry from additional factors distinguishing our two cohorts (e.g., geographic location; initiation of 17-OHPC treatment; proportion nulliparous). Thus, we should reflect carefully on notions of stability. The size of the observed effect of advancing pregnancy, which appears to have been small in the current study, might depend on the focal gestational interval and on the outset prevalence of diverse vaginal microbiota in the study population. These (likely subtle) temporal dynamics should be studied further in larger cohorts of pregnant women residing in various geographic settings. Supporting vaginal health in the periconceptional period remains an important goal.

In this study, the progression of gestation culminating in delivery was reflected in the vaginal ecosystem as a period of relative stability ending with a robust disturbance. This disturbance mainly entailed a loss of dominance among the previously dominant bacterial taxa (often but not always a *Lactobacillus* species) and a local pro-inflammatory cytokine response. In demonstrating a pervasive increase in vaginal bacterial diversity coincident with delivery, our study agrees with earlier reports, including our own[7] and those limited to a single postpartum timepoint[5,6,8,9]. (It may also extend to nonhuman primates: among wild baboons sampled at various reproductive stages, vaginal bacterial communities with the highest diversity were associated with postpartum amenorrhea[62].) As others have noted[6,8], this disturbance likely takes root in the precipitous hormonal flux of childbirth. Within days of delivery, the concentration of circulating estrogens plummets by 100- to 1000-fold[1,63,64]. At the same time, and in vaginal and cesarean deliveries alike, the vaginal epithelium thins considerably as cells become smaller, inactive, and devoid of glycogen[53]. Also influenced by hormones is the signaling and activity of immune and non-immune (e.g., epithelial) cells in ways that shape the vaginal mucosal response to infection[54]. Yet, hormonal control is not the entire story—the process of uterine involution releases an alkaline lochia into the vaginal lumen for about a month after delivery[65]. Suggestive of ongoing tissue remodeling, Nunn et al. (2021) detected higher levels of hyaluronan and Hsp70 in vaginal fluid collected four to

seven weeks after delivery compared to pregnancy-collected fluid from the same women[8]. Given that the local vaginal environment changes dramatically, it is perhaps not surprising that the immediate post-delivery microbiota appears to bear little resemblance to its pre-delivery state. The accompanying cytokine response, in this context, could be instigated by host hormonal changes[54], the presence of diverse bacteria[49], or feasibly by both. We conclude that in mirroring unavoidable developmental events, the vaginal ecosystem experiences a delivery-associated disturbance that is not likely to be prevented by modifying behavior or medical practice, but one that is part and parcel to the human life cycle—a programmed perturbation of the human microbiome. Nonetheless, for cases in which the immediate post-delivery vaginal ecosystem might be classified as optimal—just two cases in our study—we should seek to understand the basis for this resistance or high level of resilience to disturbance.

Our study suggests that the rate and pattern of postpartum recovery—defined as transitioning out of the diverse post-delivery state—varied among women and would-be dominant taxa. For example, at approximately 9.5-months postpartum (the latest time-point at which cytokines were assessed) nearly a third of participants remained in a relatively nonoptimal state with diverse microbiota and elevated cytokine levels. We suspect this finding reflects, to some degree, the considerable amount of variation that exists among women in the rate and pattern of postpartum fertility return[34,35,66]. This variation is clearly depicted by Bouchard et al. (2018), who measured hormone concentrations in daily postpartum urine samples, as ranging from relatively early ovarian activity (e.g., a month or two after delivery, while exclusively breastfeeding) to prolonged ovarian quiescence (e.g., first ovulation 11 months after delivery, on the heels of complete weaning)[67]. In turn, this variation could underpin varying degrees of postpartum vaginal atrophy[68]. In the current study, we did not measure postpartum hormone levels and our sample size was small with respect to the analysis of presumed proxies (first menses; complete weaning). Aside from host physiology, the availability of colonists might also influence recovery; indeed, little is known about the dispersal routes of vaginal *Lactobacillus* species. We found that women who initiated contraception shortly after delivery had a higher probability of transitioning to *Lactobacillus* dominance than those who never initiated, which could be driven by differences in exogenous hormone exposures and/or sexual transmission opportunities. The contraception methods used by our participants were heterogeneous and may have had disparate effects. In a recent study of postpartum Kenyan women, Whitney et al. (2021) reported that IM depot medroxyprogesterone acetate (DMPA), which can reinforce hypoestrogenism, slightly hindered *Lactobacillus* recovery relative to non-hormonal methods[69]. Further research is needed to characterize the effects of contraceptives on the postpartum vaginal ecosystem.

We also found that bacterial taxa differed in their postpartum probability of becoming dominant. *Lactobacillus iners* fared relatively well soon after delivery; indeed, it is well known as a successful transitional species, yet has an ambiguous role in human health and disease[18–20]. Over the postpartum year, many women spent time in states quasi-dominated by non-*Lactobacillus* species; for example, *Bifidobacterium*, which did not appear to provoke inflammation, is capable of acidifying the vaginal environment[52], and could signal a period of ongoing microbial exchange between mother and infant[70]. Some of these non-*Lactobacillus* dominated states might be favored under extended hypoestrogenic conditions, e.g., as seen in menopausal women[52]. We found that postpartum transitions to *L. crispatus* dominance occurred relatively rarely, and that this state remained underrepresented among pregnant women with a history of prior live birth (i.e., among the participants in our study with the most remote history of childbirth). This observation would appear to indicate a more complicated identity than "optimal" for this species, which may

have as-yet uncharacterized requirements or vulnerabilities. Understanding why nonoptimal states persist in some postpartum women will be crucial to designing appropriate interventions aimed at mitigating associated health risks. Our study is limited by a lack of direct data concerning postpartum hormonal changes, bleeding patterns, and the resumption of sexual activity. Still needed are studies that capture and control for the wide array of women's postpartum physiologies and behaviors.

Several studies have reported a reduced prevalence of *L. crispatus* dominance among pregnant women with a history of prior delivery compared to those with no such history[59,71,72]. All were cross-sectional studies of healthy participants enrolled at urban clinics (Toronto, New York City, Helsinki). Discussing their findings, Nasioudis et al. (2017) posited that differences in host immune status, known to occur between first and subsequent gestations, might play a role[71], while Freitas et al. (2017) suggested host behavior as a possible influence (e.g., more cautious during first gestations), as well as pregnancy-associated changes in host physiology, or delivery-associated changes in the vaginal microbiota, if these were to persist[59]. While not excluding the first three scenarios, our study supports the fourth (delivery-associated changes) by demonstrating in a single longitudinal cohort that *L. crispatus* dominance was common before delivery, especially among nulliparous women; that it was depleted at delivery, as were most dominated states; and that its incidence remained low throughout the year after delivery, as other states recovered. Our findings highlight the strength of the longitudinal study, in which participants serve as their own controls, to investigate associations between exposures—in this case to childbirth—and the status of the vaginal ecosystem. But questions remain—why is the shadow of delivery so long? Why does *L. crispatus* remain less likely to dominate at much later timepoints—here, throughout subsequent pregnancies? The answer could lie solely with *L. crispatus* if, for example, this species were particularly sensitive to delivery (singularly extirpated) and/or severely limited in dispersal compared to other species. Additionally, the parous host might remain inhospitable to *L. crispatus*. As alluded to above[59,71], a history of childbirth is associated with a persistent shift in host hormonal status[73–75] and immune responsiveness—the latter triggered by exposures to paternal antigen[76]. Regardless of underlying causes, one implication of these findings is that future studies should take reproductive history into account, e.g., controlling or matching for parity[71,72]. Another implication is that past studies might be reframed; for example, our SU and UAB cohorts, the latter exhibiting lower rates of nulliparity and *L. crispatus* dominance, were headlined as racially distinct[17], while they were also distinct in terms of reproductive history. Longitudinal studies characterizing the vaginal ecosystem before and after childbirth are still needed in populations at highest risk of delivering preterm (e.g., our UAB cohort) or acquiring HIV. With the goal of reducing these and other health risks (e.g., recurrence of bacterial vaginosis[77]), *L. crispatus*-based live biotherapeutic products are under active development. In the absence of other risk factors, a pregnant woman with a history of prior term birth would be considered at relatively low risk of preterm birth[78,79]. Therefore, we might consider whether alternatives such as *L. gasseri* might be adequately protective in the parous context.

Disturbance events have long been viewed as opportunities to study how ecological communities assemble in their natural environments[80]. We propose that the postpartum and interpregnancy phases represent opportune times in which to investigate the ecology of *L. crispatus* and other species—their capacity to disperse, colonize, persist, and diversify—in relation to important changes taking place in the host and local microbial community. In addition to human-population-adapted interventions, we might also consider those that are life-stage-adapted in our efforts to support vaginal health and improve outcomes for women and children.

## Methods

### Ethics statement

The study was approved by the Institutional Review Boards of Stanford University (protocol no. 21956) and the University of Alabama at Birmingham (protocol no. X121031002). All participants provided written informed consent before completing an enrollment questionnaire and providing biological samples.

### Study design

Our objective was to characterize the vaginal ecosystem's response to and recovery from childbirth. Our participants were healthy human volunteers, and the experimental design was a prospective longitudinal study of preterm birth. Individuals presenting for prenatal care or preconception consultation at the obstetrics clinics of Lucile Packard Children's Hospital at Stanford were invited to participate. Inclusion criteria were age 18 years or older, singleton gestation, not immunosuppressed, able to perform the study procedures, and able to provide written informed consent. Participants electing to continue participation after the end of a pregnancy or to resume participation at the start of a subsequent pregnancy were re-enrolled.

Demographic and clinical data, including reproductive histories, were collected in a detailed questionnaire at enrollment and in brief follow-up questionnaires at prenatal visits. Participants who had performed postpartum sampling and were therefore selected for the current study were asked to recall events such as the initiation of contraception, resumption of menstruation, and cessation of lactation in a postpartum questionnaire. We did not track hormone levels in this study. Mid-vaginal swabs were self-collected by the participants and subsequently analyzed for microbiota composition using 16S rRNA gene amplicon sequencing. Presented here for the first time are sequence data from 1,589 unique vaginal swabs representing 58 participants and 74 pregnancies, and encompassing 10 pre-gestational, 812 gestational, and 767 post-gestational timepoints. In 40 of these pregnancies, paired vaginal swabs collected ~3 weeks before, ~6 weeks after, and ~9.5 months after delivery were subjected to a human cytokine/chemokine multiplexed bead assay.

Our design also drew upon previously published samples and data. Earlier cohorts from our longitudinal study provided samples that led to the identification of a vaginal microbiota signature of preterm birth and a delivery-associated disturbance[7]. Here, we re-amplified and re-sequenced 80 of those samples, which represent three participants and four pregnancies, including 61 gestational and 19 post-gestational timepoints. These samples were selected for re-sequencing because they completed the timelines of three participants for whom additional pregnancies and/or postpartum timepoints were available, and because the original (454) sequence data were incompatible with the current study. As a result, two of the 25 deliveries evaluated in our 2015 work[7] re-appear among the 72 deliveries evaluated here.

Earlier cohorts from our longitudinal study also provided samples that led to the replication and refinement of a vaginal microbiota signature of preterm birth[17]. Because the original sequence data from these samples were compatible with the current study, we were able to merge them directly. The 2017 dataset represented longitudinal sampling of 39 pregnant participants enrolled at Stanford (893 gestational and 4 post-gestational timepoints) and 96 pregnant participants enrolled at the University of Alabama at Birmingham (1,282 gestational timepoints; participants referred for IM 17-OHPC therapy)[17]. Of these samples, 241 from SU participants were re-amplified, re-sequenced, and used to monitor for batch effects (Supplementary Fig. 9). For 40 of these pregnancies ($n = 11$ SU; $n = 29$ UAB), paired vaginal swabs collected at ~18 and ~35 weeks of gestation were subjected to a human cytokine/chemokine multiplexed bead assay.

 

Additional details about the earlier cohorts can be found in DiGiulio et al. (2015)[7] and Callahan et al. (2017)[17]. All participants were enrolled using the same study design, procedures, and methods[17]. Demographic and clinical details are provided in Supplementary Tables 1 and 2, respectively, and an overview of the study design appears in Supplementary Fig. 1.

## Sample collection

Sample collection occurred on a weekly basis from enrollment to delivery, and on a monthly basis from delivery to a maximum of 36 months postpartum or the start of the next pregnancy. Participants were asked to collect a first postpartum sample at or around six weeks after delivery. Many of our post-delivery re-enrollees elected to end participation after one year of follow-up (Supplementary Fig. 1). At each timepoint, two mid-vaginal specimens were self-collected by the participant using sterile Catch-All™ Sample Collection Swabs (Epicentre Biotechnologies) and placed at −20 °C. Following transport to the laboratory, the swabs were stored at −80 °C until further processing. The samples analyzed in this study were collected between November 2011 to September 2018.

## Measurement of cytokine/chemokine concentrations

Pregnancies were selected for cytokine/chemokine analysis if paired vaginal swabs were available within each of the designated timeframes (no other selection criteria were applied). To extract cytokines/chemokines, frozen vaginal swabs were placed in ice-cold FACS buffer and incubated overnight (-12 h) at 4 °C. The swabs were vortexed and removed from the FACS buffer, which was then centrifuged at 1,000 × g for 10 min. The Luminex assay was performed on the supernatant, in duplicate, using a custom kit of 20 analytes: IFN-γ, IL-1α, IL-1β, IL-4, IL-5, IL-6, IL-8, IL-10, IL-12p70, IL-13, IL-17, IL-21, IL-23, IP-10, I-TAC, MIG, MIP-1α, MIP-1β, MIP-3α, and TNF-α. The assay was run on a Luminex FLEXMAP 3D instrument. For each analyte, the raw data were checked for adequate correlation between duplicates, absence of plate effects, and an acceptable number of measurements falling outside the limits of detection. IL-4, IL-8, IL-12p70, and IL-13 failed one or more of these quality checks and were excluded from further study. Analyte concentrations falling below the lower limit of detection were set to half the minimum detectable concentration for the analyte. Those falling above the upper limit of detection were set to the upper limit.

## DNA extraction, 16S rRNA gene amplification, and amplicon sequencing

Genomic DNA was extracted from each vaginal swab using a PowerSoil DNA Isolation Kit (MO BIO Laboratories) according to the manufacturer's instructions, except for the inclusion of a 10-minute 65 °C incubation step immediately following the addition of reagent C1. A fragment of the 16S rRNA gene spanning the V4 variable region was PCR-amplified using primers described by Walters et al. (2016)[81]. The 75-nt forward fusion primer (5′ **AATGATACGG CGACCACCGA GATCTACACG CT**NNNNNNNN NNNNTATGGT AATTGT<u>GTGY CAGCMGCCGC GGTAA</u> 3′) contained the 32-nt 5′ Illumina adapter (shown in bold), a unique 12-nt Golay barcode (designated by the Ns), a 10-nt forward primer pad, a 2-nt forward primer linker (GT), and the 19-nt broad-range SSU rRNA gene primer 515 F (underlined). The 56-nt reverse fusion primer (5′ **CAAGCAGAAG ACGGCATACG AGAT**AGTCAG CCAGCC<u>GGAC TACNVGGGTW TCTAAT</u> 3′) contained the 24-nt 3′ Illumina adapter (reverse complemented; shown in bold), a 10-nt reverse primer pad, a 2-nt reverse primer linker (CC), and the 20-nt broad-range 16S rRNA gene primer 806 R (underlined). Amplification, purification, quantification, and pooling were carried out as described by Callahan et al. (2017)[17]. Amplification took place in triplicate 25 μl reactions using 1× HotMasterMix (5 PRIME), 0.4 μM concentrations of each primer, and 3 μL of DNA template. Thermal cycling consisted of

an initial denaturation step of 94 °C for 3 min, followed by 30 cycles of 94 °C for 45 s, 50 °C for 60 s, and 72 °C for 90 s, with a final extension step of 72 °C for 10 min. Triplicate reactions were pooled and purified using an UltraClean-htp 96-Well PCR Clean-Up Kit (MO BIO Laboratories). Amplicon DNA concentrations were assessed using a Quant-iT High-Sensitivity dsDNA Assay (Invitrogen) and normalized within pools. Amplicon pools were subjected to paired-end sequencing (2 × 250 bp) on an Illumina HiSeq 2500 instrument at the DNA Services Lab, Roy J. Carver Biotechnology Center, University of Illinois at Urbana-Champaign.

## Sequence analysis

Raw reads were demultiplexed using QIIME[82] and trimmed of non-biological sequence using cutadapt[83]. DADA2[84] was used to filter and truncate the reads, infer amplicon sequence variants (ASVs), and remove chimeras. Because ASVs are consistent labels[85], we were able to merge our dataset with the dataset of Callahan et al. (2017)[17], retrieved from the Stanford Digital Repository (https://purl.stanford.edu/yb681vm1809). All subsequent analyses were performed on the merged dataset. Taxonomy was assigned using DADA2's implementation of the RDP naïve Bayesian classifier[84,86] and a Silva reference database (version 132)[87]. ASVs were placed in a phylogenetic tree using QIIME 2's SEPP-based fragment insertion plugin[88–90] and a Silva backbone tree (version 132)[87]. The count table, taxonomy table, phylogeny, and sample-associated data (including the cytokine/chemokine data) were integrated into a single object using the phyloseq R package[91]. ASVs assigned to the domain *Bacteria* were retained. The final dataset contained 643,045,850 reads, 5,622 235-nt ASVs, 3,848 unique vaginal swabs, and 4,121 unique amplicons, of which 273 represented technical replicates. The mean (±SD) library size was 156,041 (±42,865) reads per amplicon. Thirty-four swabs with poor sequencing yield (library size <40,000 reads) were excluded from most analyses. Batch effects were not observed (Supplementary Fig. 9) and we selected the technical replicate with the highest sequencing yield for subsequent analyses.

## Lactobacillus species assignment

Using VSEARCH[92], ASVs assigned to the genus *Lactobacillus* ($n = 175$ ASVs) were queried against a database of 16S rRNA gene sequences representing *Lactobacillus* type strains in NCBI's RefSeq database ($n = 308$ strains). Hits were accepted at pairwise identities ≥98%, returning 235-nt alignments with ≤5 SNPs. ASVs were assigned to the reference species with the highest pairwise identity, with ties resulting in multi-species assignments. Using this approach, assignments were made to 169 *Lactobacillus* ASVs which together represented >99.99% of reads mapped to the 175 *Lactobacillus* ASVs in our dataset. Our method was also useful for standardizing the multi-species assignments, thus enabling downstream taxonomic agglomeration. In a phylogenetic tree inferred for the 169 *Lactobacillus* ASVs with species-level assignments, clades were consistent with the species-level labels. For brevity's sake, we have shortened some multi-species assignments to the single species known to dominate the human vaginal microbiota, namely, *L. crispatus* (also *L. acidophilus*, *L. gallinarum*), *L. gasseri* (also *L. hominis*, *L. johnsonii*, *L. taiwanensis*), and *L. jensenii* (also *L. fornicalis*). *Lactobacillus iners* was uniquely assigned. Using a similar approach, we manually curated the species-level labels of all remaining ASVs ranked in the top 100 most abundant study-wide.

## Community state designation

We adopted a simple, flexible approach to designating vaginal bacterial community states. For each sample, we (1) calculated the Shannon diversity index and (2) identified the most abundant (top) ASV. Top ASVs have three characteristics: a sequence, a taxonomy, and a frequency. In general, we relied on combinations of these features to

define states of interest. For example, samples with a Shannon diversity index > 2 or a top ASV frequency <0.5 might be labelled as diverse, while all others were labelled according to the taxonomy of their top ASV. States designated in this manner agreed well with the CSTs assigned using the clustering methods described by DiGiulio et al. (2015)[7], except that samples dominated by taxa other than the four major *Lactobacillus* species were held apart and not lumped into a heterogeneous CST IV.

## Statistical analysis

Given the nature of our study design and data, our concerns were to (1) meaningfully integrate multiple domains, namely, the vaginal 16S rRNA gene surveys and cytokine/chemokine profiles; (2) appropriately deal with the compositionality of the microbiota survey data and censoring of the time-to-event data; and (3) account for the tendency of repeated measures to be correlated. In this study, we accounted for repeated measures within the timeframe flanked by the start of a focal pregnancy and the start of a next pregnancy. If a participant contributed two such sets, and each set satisfied the analytic criteria (e.g., a requirement for >1 timepoint), then both sets were used. Statistical analyses were performed in R (version 4.0.4; https://www.r-project. org) using RStudio (version 1.4.1717; https://www.rstudio.com). Briefly, alpha- and beta-diversity metrics were calculated using the phyloseq or vegan R packages. Temporal trends in alpha- and beta-diversity were modeled using LME regression with the pregnancy as the random effect as described by DiGiulio et al. (2015)[7] (nlme::lme R function). Associations between a history of prior live birth and vaginal community states were assessed using binary logistic regression with robust (i.e., "sandwich") standard errors (stats::glm R function and sandwich R package). Postpartum time-to-event functions were estimated using Kaplan-Meier methods and evaluated for associations with predictor variables using univariate Cox regression models (survival and survminer R packages). Analysis of differential (relative) abundance was accomplished using ALDEx2[93] (ALDEx2 R package) in parallel with rank-based tests. A tree-based version of sparse discriminant analysis (a supervised method) was used to identify lineages distinguishing between late gestational and early postpartum vaginal bacterial communities[94] (treeDA R package). Finally, relationships between taxa and cytokines were explored using repeated measures correlations[95] and network visualization (rmcorr and igraph R packages) and were examined further at the late postpartum timepoint using constrained correspondence analysis (vegan::cca R function). Additional R packages used for data analysis and plotting included tidyverse, magrittr, ggtree, ggpubr, and rstatix. *P* values < 0.05 were considered significant unless otherwise noted.

## Reporting summary

Further information on research design is available in the Nature Portfolio Reporting Summary linked to this article.

# Data availability

Raw sequencing reads were deposited in NCBI's Sequence Read Archive under BioProjects PRJNA393472[17] and PRJNA821262 (this study). Source data are provided with this paper.

# Code availability

Input data and rendered R markdown files implementing the bioinformatic and statistical analyses are available at the Stanford Digital Repository (https://purl.stanford.edu/pz745bc9128).

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

## Acknowledgements

We thank the study participants, as well as Ana Campos, Zaida Esquivel, Nomfuneko Mafunda, Cele Quaintance, Hector Romero, Adrian Yabut, study coordinators at the March of Dimes Prematurity Research Center, and nursing staff in the obstetrical clinics and the labor and delivery unit of Lucile Packard Children's Hospital. We also thank Ben Callahan, Les Dethlefsen, Jessica Grembi, Stephen Popper, and members of the Relman lab for their input and feedback. This research was supported by the March of Dimes Prematurity Research Center at Stanford University and by the Bill & Melinda Gates Foundation (Grant OPP1189205 to J. Ravel, D. Relman; Grant OPP1113682 to D. Relman, D. Kwon).

## Author contributions

Conceptualization: E.K.C., D.A.R. Methodology and investigation: E.K.C., D.B.D., A.R., R.J.W., G.M.S., D.K.S., S.P.H., D.S.K., D.A.R. Formal analysis: E.K.C., L.S., S.P.H., D.S.K. Writing—original draft: E.K.C. Writing—review and editing: all authors. Supervision: D.K.S., D.S.K., D.A.R.

## Competing interests

The authors declare no competing interests.
