## [Peer Review File · Nature Communications]

REVIEWER COMMENTS

Reviewer #1 (Remarks to the Author):

o Summary:

§ This manuscript provides information about the longitudinal dynamics of the vaginal microbiota and immune state through, during, and following childbirth. The primary finding of this manuscript is a long postpartum duration of *Lactobacillus crispatus* deficiency. Secondary findings are that prior live birth was associated with lower odds of *L. crispatus* dominance during pregnancy, and this relationship was dependent upon the time since prior pregnancy.

§ The methods used in this study are:

- Amplicon sequencing of the 16S rRNA V4 region.
- Luminex assays for cytokine and chemokine concentrations.
- 82 SU subjects with 100 pregnancies,
- 96 UAB subjects w/96 pregnancies

o Issues to be addressed

§ Title: The study does not include the menstrual cycle, and so "reproductive cycle" seems misleading.

§ Line 96: "following 17 subjects over successive gestations". Isn't this 18? (100-82). Were both sets of postpartum samples from these 18 subjects used in the postpartum analysis? If so, how were repeated measures accounted for in models (random effects?)?

§ Line 290: "The prevalence of *Lactobacillus* dominance fell sharply at delivery (Fig 2c and Fig 4a)"

□ This observation does not include *L. iners* (Fig. 2c), and barely include *L. jensenii*. Be more specific here.

§ Lines 295-296: "Indeed, as postpartum diversity recovered to near pre-delivery levels (Fig 1a), the prevalence of *Lactobacillus* dominance flagged (Fig 4a)"

□ Specify this as postpartum Shannon diversity. Also, again, be specific here that it is not all *Lactobacillus* dominance but rather non-*L. iners* *Lactobacillus* dominance. The word "flagged" is not appropriate to describe this observation. Perhaps more simply "remained low".

§ Lines 300-310:

□ Please describe the model used in the methods section. Currently, the methods contain only the R package for measuring Cox proportional hazards. While the exposures are clearly listed, the outcome needs clarification (was it non-*L. iners* *Lactobacillus* dominance?) as do any confounders or

effect modifiers (for example, preterm/term delivery, induction or no induction, maternal age, race, vaginal tearing at birth) . In addition, because there were 17 subjects which were followed for the successive gestation, were repeated measures accounted for in the model?

§ Lines 346-354:

□ Similar to above, for the models here, please describe them completely in the methods. For the exposure: was it the history of any live birth (regardless of number of prior live births?). For the outcome, how is "harboring majority *L. crispatus* vaginal bacteria communities" defined? Were multiple gestational samples used? What is "majority"? Also, what confounders or effect modifiers were used here (for example, time of contraception onset post prior delivery, preterm/term prior delivery).

□ Why isn't *L. gasseri* discussed? It is also significantly associated with history of prior live birth.

§ Lines 405-410:

□ I do not understand what the authors are saying here, especially the sentence "Nonetheless, a lack of ... conflicting developmental regime." Clarify the point and, ideally, remove that specific sentence.

§ Line 428:

□ Use of the word "memory" is inappropriate personification of the system. "Resemblance" is better.

§ Lines 440-442: "We suspect this finding [variations in postpartum microbiota transitions] reflects, to some degree, the considerable amount of variation that exists among women in the rate and pattern of postpartum fertility return."

□ I agree with this statement, but it excludes consideration of other major factors occurring at delivery that could significantly impact the return of the original microbiota state (though, admittedly, we do yet not know the specific effects of most factors on postpartum microbiota). This includes the use of chemical or physical induction mechanisms during labor, maternal age at delivery (participants range from 25-43 yo which represent vastly different stages in reproductive years), vaginal injury during birth which may require the use of sutures and subsequent vaginal cleaning such as saline baths, etc, which undoubtedly impact the vaginal environment), and breastfeeding throughout the postpartum time (not just cessation of breastfeeding which implies all mothers breastfed).

§ Line 495 requires a reference for the statement that prior term birth is not a risk factor for preterm birth.

§ Line 543: it is unclear how Extended Figure 2c assesses compositional variation associated with technical replication. Showing gray points as technical replicates does on this graph does not determine concordance of composition between replicates.

§ Line 545-550: It is worthwhile to at least briefly state the study design procedures and methods from references 7 and 17 here, especially differences, as it may impact interpretation of the results of this study.

§ Figures 3 and 4 and Extended Data 3, 4, 7, and 8 need species and genus names italicized.

Reviewer #2 (Remarks to the Author):

This study reports the novel and interesting finding that parturition causes abrupt changes in the vaginal microbiome, shifting it towards a more diverse composition. This shift is also associated with a strong host immune response (as seen quite obviously in Extended Data Figure 5a), and seems to be reversible (at least in some women), but within a variable time-scale. This study provides compelling arguments to factor post-partum follow up and sample collection in vaginal microbiome studies, and adds biological knowledge that can help contextualize known associations between inter-pregnancy intervals and risk of adverse pregnancy outcomes. Despite not exploring this in detail, it's reasonable to say that these findings support the importance of targeting vaginal microbiome parameters also after pregnancy to optimise reproductive health. The figures are clear and of high quality, and the discussion is well written and well referenced, including previous work in the field with relevant observations and the connections with the authors previous work on the same cohort. The microbiome and cytokine measurement methods used are standard in the field, and longitudinal characteristics of the dataset were appropriately taken into consideration when performing the statistical analyses. Data and source code for the statistical analysis was made publicly available and the STORM guidelines for microbiome studies were adhered to. One of the weakest points of the work is the disconnect between the SU (from where most of the key findings are taken) and the UAB cohorts but I consider that the authors acknowledged the limitations of performing comparisons between these cohorts satisfactorily throughout the text. Ideally, another cohort with post-partum sampling should be used to provide a more direct replication. Nevertheless, I find that the key messages are well supported by the data.

Some minor comments:

- What is responsible for the large proportion of variance explained (Conditional $R^2 > 0.6$) in the linear mixed effect models accompanying Figure 1a, the random slope or the random intercept components? It would be good to report the variance explained by each random effect (e.g. checking the CIs for the random effects with the confint function or by comparing the conditional R^2 measures for the ri only vs the rs + ri model) to better support biological interpretation.

- An observation I find interesting is that there are more "CSTIV-C0"-like compositions, with enrichment of Prevotella and other anaerobes. I think if possible, it would be good to test formally if the high-diversity microbial compositions or "CST IV subtypes" found during gestation and post-partum are different.

- Are some of the key correlations seen in the CCA plots of Figure 3c visible using only pre-partum samples? For example, can we detect an association between Prevotella and IL-10 or IL-23 in pre-partum samples only?

Reviewer #3 (Remarks to the Author):

Dear Authors,

This is a remarkable study, highlighting the significance of longitudinal vaginal sampling. I would like to congratulate you for the efforts conducting the research and the interesting results.

There is one issue I would like to point out; one of your interesting findings is that delivery-associated non-optimal features persisted into the tenth postpartum month. In regard to postpartum women, this can be influenced from temporary loss of estrogen, bleeding (especially at the first 4-6 weeks postpartum) and resumption of sexual intercourse. However, the data regarding these three components are not defined in the manuscript (you mention the hormonal aspect, but I could not find the actual numbers and for how many women you had these data).

1. Hormonal changes: there is a wide spectrum of postpartum hormonal “phenotypes”- some women resume menstruation shortly after delivery albeit breastfeeding, some after reduction of breastfeeding frequency and others only after waning lactation completely. Also, many breastfeeding women do not have vaginal atrophy at all although they breastfeed and amenorrheic. This is similar to the clinical variability seen among menopausal women. In clinical practice, this may be associated with varying degrees of vaginal atrophy which can resolve spontaneously shortly after delivery or persist as long as lactation continues (I do not know a study that described this clinical spectrum, but see: <https://pubmed.ncbi.nlm.nih.gov/32604213/> and <https://pubmed.ncbi.nlm.nih.gov/32569019/>).

Vaginal atrophy is therefore unpredictable in the postpartum period and can further determine vaginal microbiome. Also, some women continue to breastfeed until their next pregnancy (including in the beginning of their pregnancy) which may further influence the vaginal hormonal status and microbiome.

2. Bleeding: prolonged bleeding in the early postpartum can last 4-6 weeks. There are data showing that blood has a significant influence on the diversity of the vaginal microbiome. It was shown in previous longitudinal studies that menstruation cause temporal changes (<https://pubmed.ncbi.nlm.nih.gov/22553250/>). So, the significant change found following delivery in the first PP sample, can result from ongoing bleeding at the time of sampling, which can affect inflammatory environment as well (including the possibility of actually sampling uterine lochia and not vaginal discharge).

This can be relevant during pregnancy as well in case women experience bleeding during pregnancy - I do not know if you have the data, but it is interesting whether bleeding during pregnancy influences the vaginal microbiome.

3. Resumption of sexual intercourse- you write it in the discussion but you do not show data.

These parameters may not influence your main findings, but I think they should be acknowledged as limitations of the current data.

- The last paragraph in the results section seems to belong to the Discussion section.

- Fig 4e- I could not find P2.

- Extended data Fig 7-“SU-post” refers to the first PP sample?

Thank you,

Ahinoam Lev Sagie

Point-by-point responses to reviewer comments for manuscript # NCOMMS-22-48924-T

We thank the reviewers for their thoughtful and constructive feedback. Point-by-point responses to each reviewer's comments and suggestions are provided below. We have also revised our manuscript and addressed the questions and concerns raised by the reviewers. We believe that these revisions have strengthened our work and improved its quality and clarity. Again, we thank the reviewers for their time and effort in helping us to enhance our manuscript.

Reviewer comments are italicized; author responses are in bold; line numbers refer to the original unrevised text.

Reviewer #1

*Summary: This manuscript provides information about the longitudinal dynamics of the vaginal microbiota and immune state through, during, and following childbirth. The primary finding of this manuscript is a long postpartum duration of *Lactobacillus crispatus* deficiency. Secondary findings are that prior live birth was associated with lower odds of *L. crispatus* dominance during pregnancy, and this relationship was dependent upon the time since prior pregnancy.*

The methods used in this study are:

Amplicon sequencing of the 16S rRNA V4 region

Luminex assays for cytokine and chemokine concentrations

82 SU subjects with 100 pregnancies

96 UAB subjects with 96 pregnancies

Issues to be addressed:

(1) *Title: The study does not include the menstrual cycle, and so "reproductive cycle" seems misleading.*

We have removed the phrase "reproductive cycle" and revised the title to, "Abrupt perturbation and delayed recovery of the vaginal ecosystem following childbirth"

(2) *Line 96: "following 17 subjects over successive gestations" Isn't this 18? (100-82). Were both sets of postpartum samples from these 18 subjects used in the postpartum analysis? If so, how were repeated measures accounted for in models (random effects?)?*

The number of subjects followed over successive gestations was 17, rather than 18, because one subject was followed over three gestations. (This subject's second enrolled gestation and ensuing interpregnancy interval

are the focus of Figure 4e. The subject's first enrolled gestation ended in miscarriage.) Therefore, the study included: 1 subject (x3 pregnancies) + 16 subjects (x2 pregnancies) + 65 subjects (x1 pregnancy) = 82 subjects (100 pregnancies) from the SU cohort. We've inserted the following sentence at line 116:

"Sixteen SU subjects were followed over two pregnancies, and one was followed over three."

The number of subjects with two sets of postpartum samples was 13, rather than 17, because several births were not followed up with postpartum sampling, and because we did not consider samples collected after a miscarriage as a "postpartum set" (Extended Data Figure 1). (Our postpartum dataset included 59 subjects, as shown in Table S1, with 72 deliveries, shown in Table S2.) For the postpartum analyses, both sets of postpartum samples were used, as long as both sets satisfied any criteria specific to the analysis (e.g., for the postpartum linear mixed-effects (LME) model portrayed in Figure 1a, we required the postpartum set to have at least two samples, which reduced the number of subjects with two postpartum sets from 13 to nine in an analysis of 65 postpartum sets from 56 subjects).

We accounted for repeated measures within pregnancies (e.g., the random effect specified in our LME model was a variable called "PregID"). As defined in our study, a given PregID would apply to any sample collected from the conception of the focal gestation to the conception of a next gestation. We considered the PregID as the experimental unit of the study. In other words, samples from the same subject (SubjID) but from different pregnancies (PregIDs) were not considered repeated measures. We've added the following two sentences to the Methods section at line 648:

"In this study, we accounted for repeated measures within the timeframe flanked by the start of a focal pregnancy and the start of a next pregnancy. If a subject contributed two such sets, and each set satisfied the analytic criteria (e.g., a requirement for > 1 timepoint), then both sets were used."

The LME models underpinning Figure 1a are specified in our analysis code (file "2-fig1-post.html"). We've added a **README** file that clarifies which analyses are found in which code files. All code, data, and readme files can be found at the Stanford Digital Repository at the following permanent URL: <https://purl.stanford.edu/pz745bc9128>.

Our decision to consider the pregnancy (including any samples collected after delivery) as the experimental unit was motivated by several factors. First, to say upfront, when we account for repeated measures within subjects (e.g., changing the random effect specified in our LME models from the pregnancy ID to the subject ID), we obtain results that are congruent with, and nearly identical to the results presented in our manuscript, as shown in Figure 1 below. Indeed, subjects with two pregnancies made up a small fraction of

the overall study cohort. Second, we found that random effects slopes of postpartum sets from the same subject were not strongly correlated (Pearson's $r = 0.41$; $P = 0.27$; $n = 9$ pairs). Third, postpartum sets from the same subject were initiated (first sample collected) an average of 636 days apart (range 392-1196 days apart), and it is generally understood that within-subject compositional dissimilarity increases with increasing sampling interval (as shown in Extended Data Figure 2c, for example). Finally, our focus was on the events and circumstances surrounding the delivery and extended postpartum phase, which we found could differ substantively from birth to birth in the same woman (for example, differences in labor onset type, antibiotic exposure, and delivery mode).

Figure 1. Alpha diversity, measured using the Shannon diversity index, is plotted against day relative to the end of gestation (vertical dashed line). For each point ($n = 3,848$ unique vaginal swabs), 16S rRNA genes were amplified, sequenced, and resolved to amplicon sequence variants (ASVs). For each non-gray point, a paired swab was analyzed for cytokine/chemokine concentrations ($n = 198$ unique vaginal swabs). The data are faceted by cohort (left, UAB cohort; right, SU cohort). Solid lines depict linear mixed-effects regressions and shaded areas represent 95% confidence intervals for the fixed effect term (i.e., day). Three datasets were modeled, each limited to pregnancies ending in delivery with >1 timepoint available for analysis. P values correspond to the estimated slopes of the fixed effect term. The rate of postpartum recovery was also explored using local smoothing (loess fit; dot-dashed line). In **panel a**, **per-pregnancy** slopes and intercepts were modeled as random effects. This panel appears as Figure 1a in our paper (UAB before, $n = 93$; SU before, $n = 98$; SU after, $n = 65$). In **panel b**, **per-subject** slopes and intercepts were modeled as random effects (UAB before, $n = 93$; SU before, $n = 82$; SU after, $n = 56$).

(3) Line 290: "The prevalence of *Lactobacillus* dominance fell sharply at delivery (Fig 2c and Fig 4a)" This observation does not include *L. iners* (Fig. 2c), and barely include *L. jensenii*. Be more specific here.

Apologies. We did not clearly describe our aim, which was to introduce a set of results pertaining to the genus *Lactobacillus*, measured as dominance by any ASV assigned to this genus. (We go on to address individual species in the paragraph beginning on line 315.) We hope that the following edit more clearly signals our

intention to remain general in this and the following paragraph. We've removed the reference to Fig. 2c, because it displays species in separate panels and thus confuses the issue, and edited line 290 to the following:

"Delivery was associated with a sharp decline in the prevalence of communities dominated by a *Lactobacillus* ASV (i.e., any ASV assigned to the genus *Lactobacillus*; Fig. 4a)."

(4) Lines 295-296: "Indeed, as postpartum diversity recovered to near pre-delivery levels (Fig 1a), the prevalence of *Lactobacillus* dominance flagged (Fig 4a)" Specify this as postpartum Shannon diversity. Also, again, be specific here that it is not all *Lactobacillus* dominance but rather non-*L. iners* *Lactobacillus* dominance. The word "flagged" is not appropriate to describe this observation. Perhaps more simply "remained low".

Regarding *Shannon* (line 295) and *flagged* (line 296), we have incorporated the reviewer's edits as suggested. As stated on line 293 ("*Lactobacillus* dominance by any species") and in the response to comment 3 above, we are reporting on any *Lactobacillus* dominance. We've edited line 296 to include this full phrase as used on line 293.

We should point out that these dynamics were assessed in at-risk cases, defined as cases in which the first postpartum sample did not feature a *Lactobacillus* ASV at > 0.7 frequency (and at least one follow-up sample was available for analysis). As such, cases dominated by an *L. iners* ASV at the first postpartum sample – the cases shown in Figure 2c – were excluded from this analysis because they were already in the state to which we were quantifying transitions. Our definition of at-risk appears in the Figure 4 caption.

(5) Lines 300-310: Please describe the model used in the methods section. Currently, the methods contain only the R package for measuring Cox proportional hazards. While the exposures are clearly listed, the outcome needs clarification (was it non-*L. iners* *Lactobacillus* dominance?) as do any confounders or effect modifiers (for example, preterm/term delivery, induction or no induction, maternal age, race, vaginal tearing at birth). In addition, because there were 17 subjects which were followed for the successive gestation, were repeated measures accounted for in the model?

These were univariate Cox regression models in which the response variable was the same time-to-event (survival) object as discussed in the prior paragraph – that is, a time-to-event object in which time was measured in postpartum days and the status was either event (1 = any *Lactobacillus* ASV achieves a frequency > 0.7) or right censor (0 = last day of postpartum sampling for any case in which the event was not observed). We have added these details to the Results section at line 302 and to the Methods section at line 655. The time-to-event object and Cox regression models are specified in our analysis code (file "11-edfig8-post.html").

Our models did not include any confounders or effect modifiers. The variables preterm/term (gestational day of delivery), induction or no induction (labor onset type), and maternal age (at delivery) were assessed in univariate models and reported as non-significant results on lines 309-311. The same is true for delivery mode (line 310), which is our closest approximation of vaginal tissue trauma (among those who delivered vaginally, we do not have data on vaginal tearing at birth). We did not test for an effect of race, as this subset of the SU cohort was not particularly well balanced for this factor.

As discussed in response to comment 2, we considered the pregnancy (including the delivery and postpartum period) as the experimental unit of study. In addition to the rationale presented above, we would add here that the variables (predictors) examined in our Cox regression models were only those that can vary across different pregnancies from the same person.

(6) Lines 346-354: Similar to above, for the models here, please describe them completely in the methods. For the exposure: was it the history of any live birth (regardless of number of prior live births?). For the outcome, how is "harboring majority *L. crispatus* vaginal bacteria communities" defined? Were multiple gestational samples used? What is "majority"? Also, what confounders or effect modifiers were used here (for example, time of contraception onset post prior delivery, preterm/term prior delivery). Why isn't *L. gasseri* discussed? It is also significantly associated with history of prior live birth.

The results reported on lines 346-354 pertain to Figure 5a. At present, many of the details requested by the reviewer appear in the Figure 5a caption, which we have reproduced below. In general, it is our belief that figure captions serve as an appropriate location for such details – as an alternative to the Methods section. We understand that the journal may require us to shorten our figure captions (owing to space considerations), in which case we would move this text to the Methods section. But here, we are unclear as to whether the reviewer is asking us to repeat the model description in the Methods section, or to provide it at least once.

Figure 5a caption, as currently written:

"Associations between vaginal bacterial community states (outcomes) and a history of prior live birth (exposure) in $n = 98$ SU gestations ($n = 53$ with a history of prior live birth). Six states were defined using features of the top ASV (majority status and taxonomic identity) and modeled separately as dichotomous outcomes (one-versus-rest). Wald 95% CIs were calculated using a clustered covariance matrix which accounts for correlation among longitudinal samples. Differences in sampling effort were addressed via repeated (100 iterations) random draws of six samples per gestation. Similar results were obtained for crude (shown) and maternal age- and race-adjusted models. Results for the uncommon state "*L. jensenii*" (top ASV frequency > 0.5 & identified as *L. jensenii*) were unstable and are not shown."

The reviewer is correct – in this model, the exposure was a history of any live birth (regardless of the number), coded as 0 (no history; 0 live births) or 1 (yes history; > 0 live births). We have edited line 347 to clarify this point. Further details may be found in analysis code (file “12-fig5-post.html”), where these models are specified.

"Majority" is more than half.

Finally, we thank the reviewer for the opportunity to re-consider *L. gasseri* in the context of Figure 5a. Our hesitation in emphasizing this result concerns the issue of symmetry – that is, the odds ratio (OR) for majority-*L. gasseri* was not as high as the OR for majority-*L. crispatus* was low. Instead, it seemed closer to the truth that there was some flow from the majority-*L. crispatus* state to all other states shown in Figure 5a (all had ORs > 1). That said, we think it would be reasonable to acknowledge that the most significant "flow" was to the majority-*L. gasseri* state. We've inserted the following sentence at line 349:

"For all other states, the odds were higher – the topmost being for the majority-*L. gasseri* state (Fig. 5a)."

(7) Lines 405-410: *I do not understand what the authors are saying here, especially the sentence "Nonetheless, a lack of ... conflicting developmental regime." Clarify the point and, ideally, remove that specific sentence.*

We have removed the specific sentence and replaced it with the following sentence at line 407:

"These (likely subtle) temporal dynamics should be studied further in larger cohorts of pregnant women residing in various geographic settings."

(8) Line 428: *Use of the word "memory" is inappropriate personification of the system. "Resemblance" is better.*

We have incorporated the reviewer's suggested edit at line 428.

(9) Lines 440-442: *"We suspect this finding [variations in postpartum microbiota transitions] reflects, to some degree, the considerable amount of variation that exists among women in the rate and pattern of postpartum fertility return." I agree with this statement, but it excludes consideration of other major factors occurring at delivery that could significantly impact the return of the original microbiota state (though, admittedly, we do not yet know the specific effects of most factors on postpartum microbiota). This includes the use of chemical or physical induction mechanisms during labor, maternal age at delivery (participants range from 25-43 yo which represent vastly different stages in reproductive years), vaginal injury during birth which may require the use of sutures and*

subsequent vaginal cleaning such as saline baths, etc, which undoubtedly impact the vaginal environment), and breastfeeding throughout the postpartum time (not just cessation of breastfeeding which implies all mothers breastfed).

To the degree that we did consider these or related factors – e.g., maternal age at delivery, labor onset type, delivery mode, intrapartum antibiotics (among others) – we found minimal impact on the delivery-associated disturbance (e.g., it occurs in vaginal and cesarean deliveries alike), or on the postpartum rate of attaining *Lactobacillus* dominance. We certainly acknowledge that a larger study would be required to detect more subtle effects, and a larger study might also enable the examination of additional microbiome-oriented response variables. We consider variation in the lactation trajectory, inclusive of women who do and do not breastfeed, to be one of several factors likely underpinning variation in the rate and pattern of postpartum fertility return. Admittedly, the term "fertility return" is suboptimal, i.e., although the phenomenon to which it refers subsumes many biological factors and marks a certain culmination of processes, "fertility" per se may not be the most relevant feature in a woman's experience.

(10) *Line 495 requires a reference for the statement that prior term birth is not a risk factor for preterm birth.*

We have edited the statement on line 495 and provided two references as follows:

"Controlling for other risk factors, a prior *term* birth is associated with a relatively low risk of preterm birth (Ananth et al., 2007; Koullali et al., 2020)."

(11) *Line 543: It is unclear how Extended Figure 2c assesses compositional variation associated with technical replication. Showing gray points as technical replicates on this graph does not determine concordance of composition between replicates.*

We have deleted the phrase and reference to Extended Data Figure 2c at lines 542-543 of the Methods section. We have also clarified in the Extended Data Figure 2c caption that “points at Lag = 0 represent pairs of technical replicates”. We found it reassuring that the amount of compositional variation observed between pairs of technical replicates (assessed via Bray-Curtis dissimilarity) was among the lowest in the study, even when compared to the average level of compositional variation observed between pairs of samples collected from the same person on the order of one week apart (points at Lag = 1), the smallest interval we sampled.

(12) *Line 545-550: It is worthwhile to at least briefly state the study design procedures and methods from references 7 and 17 here, especially differences, as it may impact interpretation of the results of this study.*

This was the aim of the four paragraphs (lines 509-545) preceding this one (lines 547-550), which make up the bulk of the segment entitled "Study design". In brief, this was a long-running study that included both the SU and UAB cohorts, in which every effort was made to use consistent procedures and methods – in the clinic, in the lab, and in the processing of the sequence data. We are unaware of any detail not already mentioned that might impact on the interpretation of the results of this study. Furthermore, all of the subjects and pregnancies from references 7 and 17 appear in Supplementary Tables 1 and 2 (demographic and clinical characteristics) and in Extended Data Figure 1, which gives an overview of the study design and sampling timelines.

(13) *Figures 3 and 4 and Extended Data 3, 4, 7, and 8 need species and genus names italicized.*

Thank you, we have made these changes.

Reviewer #2

This study reports the novel and interesting finding that parturition causes abrupt changes in the vaginal microbiome, shifting it towards a more diverse composition. This shift is also associated with a strong host immune response (as seen quite obviously in Extended Data Figure 5a), and seems to be reversible (at least in some women), but within a variable time-scale. This study provides compelling arguments to factor post-partum follow up and sample collection in vaginal microbiome studies, and adds biological knowledge that can help contextualize known associations between inter-pregnancy intervals and risk of adverse pregnancy outcomes. Despite not exploring this in detail, it's reasonable to say that these findings support the importance of targeting vaginal microbiome parameters also after pregnancy to optimise reproductive health. The figures are clear and of high quality, and the discussion is well written and well referenced, including previous work in the field with relevant observations and the connections with the authors previous work on the same cohort. The microbiome and cytokine measurement methods used are standard in the field, and longitudinal characteristics of the dataset were appropriately taken into consideration when performing the statistical analyses. Data and source code for the statistical analysis was made publicly available and the STORM guidelines for microbiome studies were adhered to. One of the weakest points of the work is the disconnect between the SU (from where most of the key findings are taken) and the UAB cohorts but I consider that the authors acknowledged the limitations of performing comparisons between these cohorts satisfactorily throughout the text. Ideally, another cohort with post-partum sampling should be used to provide a more direct replication. Nevertheless, I find that the key messages are well supported by the data.

Some minor comments:

(1) *What is responsible for the large proportion of variance explained (Conditional $R^2 > 0.6$) in the liner mixed effect models accompanying Figure 1a, the random slope or the random intercept components? It would be good to report the variance explained by each random effect (e.g. checking the CIs for the random effects with the `confint` function or by comparing the conditional R^2 measures for the *ri* only vs the *rs + ri* model) to better support biological interpretation.*

For each of the three models accompanying Figure 1a, the conditional R^2 value for a version of the model with a random intercept only ("ri only model") was nearly as high as the conditional R^2 value for the Figure 1a model, which had a random intercept plus a random slope ("ri + rs model"). These observations reinforce our intuition that much of the unexplained variation in the Shannon diversity index is associated with differences between pregnancies – a statement we have added at line 1051, where conditional R^2 values are first mentioned – and perhaps especially with differences between pregnancies in what might be considered the "set-point" or baseline (i.e., the feature captured by the random intercept).

(2) *An observation I find interesting is that there are more "CSTIV-C0"-like compositions, with enrichment of *Prevotella* and other anaerobes. I think if possible, it would be good to test formally if the high-diversity microbial compositions or "CST IV subtypes" found during gestation and post-partum are different.*

Yes, we agree! Thinking along these lines motivated the analyses presented in Extended Data Figure 6 (especially panel c) and the results reported on lines 252-257 ("... nor did delivery usher in a completely novel set of diverse taxa"). In short, we did not find strong evidence for a novel diverse state associated with the postpartum phase. Although formal classification into "CST IV subtypes" is beyond the scope of this project, we would predict that delivery is associated with an increase in the prevalence of (certain) CST IV subtypes, but that the addition of postpartum data to a re-evaluation (re-classification) of subtypes would not result in the identification of an entirely new subtype.

A formal test is challenging for a few reasons. We defined "high diversity" at a cutoff of Shannon diversity index (SDI) > 2 . When we did this, "diverse" gestational samples still had higher levels of *Lactobacillus*, on average, than did "diverse" postpartum ones. However, this difference wasn't really our interest here, as we had already established that delivery was extraordinarily effective at knocking down *Lactobacillus*. Then, if we added a *Lactobacillus* filter to our definition of "high diversity" (say, SDI > 2 & *Lactobacillus* < 0.1), we no longer had enough gestational samples to do a robust analysis. So instead, we focused on community membership, using distance metrics that are presence/absence-based (at least these were insensitive to differences in the relative abundance of *Lactobacillus*). Extended Data Figure 6c presents an ordination based on such a distance metric (binary Jaccard; with similar results obtained using unweighted Unifrac), showing that diverse postpartum samples primarily clustered with diverse gestational samples from the same

subjects/cohort – the SU cohort (and not with diverse gestational samples from a different cohort – the UAB cohort).

Comparing the memberships of "diverse" (SDI > 2) vaginal bacterial communities sampled before delivery (last four gestational months) to those sampled after delivery (first four postpartum months) within the set of pregnancies that conducted postpartum sampling, such that all pregnancies in the analysis included pre-delivery and post-delivery samples, we found little difference using either the unweighted UniFrac distance (dispersion test, $P = 0.59$; location test, $P = 0.11$) or the binary Jaccard distance (dispersion test, $P = 0.87$; location test, $P = 0.04$; see Figure 2 below). However, given the challenges outlined above, we considered these results to be somewhat too dependent on arbitrary cutoffs to report in the paper.

Figure 2. Ordination of "diverse" (SDI > 2) vaginal bacterial communities sampled before delivery (black circles) or after delivery (red triangles) from the set of SU pregnancies sampled in both phases, defined here as the four months pre-delivery or the four months post-delivery. The labels at the center of the plot indicate the group centroids. The distance metric is binary Jaccard, and the ordination method is Principal Coordinates Analysis (PCoA). The first two axes are plotted.

(3) Are some of the key correlations seen in the CCA plots of Figure 3c visible using only pre-partum samples? For example, can we detect an association between *Prevotella* and IL-10 or IL-23 in pre-partum samples only?

This question came up for us, too. However, we found that we could not address it with the data in hand. At the late gestational timepoint (pre-partum samples), there were few (if any?) pregnancies in this state (e.g., high *Prevotella*) – the samples were largely unperturbed, and *Lactobacillus* dominated, so there was not enough variation among pregnancies to discern any such association. The same phenomenon occurred at the early postpartum timepoint, except at this timepoint nearly all were in the perturbed state. It was only at the late postpartum timepoint, when some had recovered, and others had not, that there was enough (well-represented) variation between pregnancies to visualize any association. We agree that it would be important and interesting to test for such associations in additional, larger cohorts, at various stages of the lifecycle.

Reviewer #3

Dear Authors, This is a remarkable study, highlighting the significance of longitudinal vaginal sampling. I would like to congratulate you for the efforts conducting the research and the interesting results. There is one issue I would like to point out; one of your interesting findings is that delivery-associated non-optimal features persisted

into the tenth postpartum month. In regard to postpartum women, this can be influenced from temporary loss of estrogen, bleeding (especially at the first 4-6 weeks postpartum) and resumption of sexual intercourse. However, the data regarding these three components are not defined in the manuscript (you mention the hormonal aspect, but I could not find the actual numbers and for how many women you had these data).

[1] Hormonal changes - there is a wide spectrum of postpartum hormonal “phenotypes”- some women resume menstruation shortly after delivery albeit breastfeeding, some after reduction of breastfeeding frequency and others only after waning lactation completely. Also, many breastfeeding women do not have vaginal atrophy at all although they breastfeed and amenorrheic. This is similar to the clinical variability seen among menopausal women. In clinical practice, this may be associated with varying degrees of vaginal atrophy which can resolve spontaneously shortly after delivery or persist as long as lactation continues (I do not know a study that described this clinical spectrum, but see: <https://pubmed.ncbi.nlm.nih.gov/32604213/> and <https://pubmed.ncbi.nlm.nih.gov/32569019/>). Vaginal atrophy is therefore unpredictable in the postpartum period and can further determine vaginal microbiome. Also, some women continue to breastfeed until their next pregnancy (including in the beginning of their pregnancy) which may further influence the vaginal hormonal status and microbiome.

We did not track hormone levels in this study. As such, our Discussion is limited to what is generally known from the literature on pregnancy and delivery, and what we might infer (albeit poorly) from our data on postpartum contraception, menstruation, and lactation. We have added a sentence to the Methods section at line 520 stating that we did not measure hormone levels. This is also stated in the Discussion (line 445).

The wide variety of postpartum hormonal trajectories is truly remarkable. We cite a paper (Bouchard et al., 2018) that, we hope, gives a sense of this variation, but we also do not know of a paper that fully describes this spectrum (might there be clusters of trajectory types?) or addresses the potential underlying drivers (might there be environmental cues yet to be identified?). Thank you for sharing your insight on the varying degree of postpartum vaginal atrophy – we have added a mention of this in the Discussion at line 445, citing the second reference listed above (Lev-Sagie et al., 2020).

[2] Bleeding - prolonged bleeding in the early postpartum can last 4-6 weeks. There are data showing that blood has a significant influence on the diversity of the vaginal microbiome. It was shown in previous longitudinal studies that menstruation cause temporal changes (<https://pubmed.ncbi.nlm.nih.gov/22553250/>). So, the significant change found following delivery in the first PP sample, can result from ongoing bleeding at the time of sampling, which can affect inflammatory environment as well (including the possibility of actually sampling uterine lochia and not vaginal discharge). This can be relevant during pregnancy as well in case women experience bleeding during

pregnancy - I do not know if you have the data, but it is interesting whether bleeding during pregnancy influences the vaginal microbiome.

The subjects in our study did not record or otherwise track bleeding in a detailed manner, but we do have data on whether blood was visible on the swab at the time of processing. (These data were available for the swabs that were processed for the current study; they were not available for the swabs that were processed for our earlier work – Callahan et al., 2017). As shown in the Figure 3 below, blood was visible on a small number of swabs (panel a), but the presence of blood was not associated with an enhancement of the delivery-associated increase in vaginal bacterial diversity (panel b). We acknowledge in the paper at lines 76 and 424 that the vaginal ecosystem likely reflects local environmental inputs, specifically, lochial discharge in the immediate wake of delivery.

Figure 3. Panel a depicts the number of swabs on which blood was visible at the time of processing – a rough proxy for vaginal bleeding at the time of self-sampling. Panel b displays the same data as plotted in manuscript Figure 2a, but with the first postpartum (firstPost) samples split on whether blood was visible on the swab at the time of processing. ****, $P < 0.0001$; ns, $P > 0.05$, Wilcoxon rank sum tests.

(3) *Resumption of sexual intercourse - you write it in the discussion, but you do not show data. These parameters may not influence your main findings, but I think they should be acknowledged as limitations of the current data.*

We agree. We have added a statement to the Discussion at line 467 acknowledging that our study is limited by a lack of direct data concerning postpartum hormonal changes, bleeding patterns, and the resumption of sexual activity.

(4) *The last paragraph in the results section seems to belong to the Discussion section.*

The goal of this paragraph was to summarize and contextualize (with respect to our earlier findings) the last segment of our Results section, specifically, the findings presented in Figure 5. These findings relate to the longest view we take on the dynamics at play in our study, coming full circle with respect to the reproductive cycle. While acknowledging that it bridges styles to a certain extent (i.e., Results and Discussion), respectfully, we would opt to maintain the paragraph at its current location, at the end of the Results section as a transition to the next.

5 Fig 4e - I could not find P2.

Thank you for pointing this out. The subject's second enrolled pregnancy (P2) is the figure's focal pregnancy. To clarify the meaning of the facet labels and explain their relationship to the x-axis (study day), we have rewritten the Figure 4e caption as follows starting at line 990:

"Temporal dynamics in an individual subject. Heatmap displaying the \log_{10} -transformed frequencies of the most abundant taxa in samples collected throughout pregnancy (1st through 3rd trimesters) and the subsequent interpregnancy interval (Postpartum years 1 and 2). This was the subject's second enrolled pregnancy. Also displayed are the last three consecutive samples (...P1) collected before the start of the focal pregnancy and the first three consecutive samples (P3...) collected after the start of the next pregnancy. The timeseries depicts two transitions to *L. crispatus*-dominance. The second was sustained until the delivery of the next pregnancy (data not shown). Study day is relative to the subject's first sample, which was collected prior to the start of her first enrolled pregnancy (data not shown), which ended in miscarriage. Prior to \log_{10} -transformation, frequencies < 0.001 were set to 0.001."

6 Extended data Fig 7 - "SU-post" refers to the first PP sample?

In Extended Data Figure 7, "SU.post" refers to all PP (postpartum) samples, i.e., any sample collected after delivery, regardless of how long after delivery. At line 1195, we have revised the Extended Data Figure 7 caption to include the following sentence:

"The Figure depicts all samples collected from pregnancies in the postpartum subset ($n = 72$ SU pregnancies), with gestational samples displayed in left facets (SU.gest; $n = 1195$ unique samples) and postpartum samples displayed in right facets (SU.post; $n = 745$ unique samples)."

REVIEWERS' COMMENTS

Reviewer #2 (Remarks to the Author):

I consider that the authors addressed my comments thoroughly and adequately.

Point-by-point responses to reviewer comments for manuscript # NCOMMS-22-48924A

Reviewer #2

I consider that the authors addressed my comments thoroughly and adequately.

We thank the reviewer for the positive feedback and for taking the time to review our work.